# Bioinformatic characterization of angiotensin-converting enzyme 2, the entry receptor for SARS-CoV-2

**Harlan Barker, Seppo Parkkila** *

Faculty of Medicine and Health Technology, Tampere University and Fimlab Ltd, Tampere University Hospital, Tampere, Finland

* seppo.parkkila@tuni.fi

## Abstract

The World Health Organization declared the COVID-19 epidemic a public health emergency of international concern on March 11th, 2020, and the pandemic is rapidly spreading worldwide. COVID-19 is caused by a novel coronavirus SARS-CoV-2, which enters human target cells via angiotensin converting enzyme 2 (ACE2). We used a number of bioinformatics tools to computationally characterize ACE2 by determining its cell-specific expression in trachea, lung, and small intestine, derive its putative functions, and predict transcriptional regulation. The small intestine expressed higher levels of *ACE2* mRNA than any other organ. By immunohistochemistry, duodenum, kidney and testis showed strong signals, whereas the signal was weak in the respiratory tract. Single cell RNA-Seq data from trachea indicated positive signals along the respiratory tract in key protective cell types including club, goblet, proliferating, and ciliary epithelial cells; while in lung the ratio of *ACE2*-expressing cells was low in all cell types (<2.6%), but was highest in vascular endothelial and goblet cells. Gene ontology analysis suggested that, besides its classical role in the renin-angiotensin system, ACE2 may be functionally associated with angiogenesis/blood vessel morphogenesis. Using a novel tool for the prediction of transcription factor binding sites we identified several putative binding sites within two tissue-specific promoters of the *ACE2* gene as well as a new putative short form of *ACE2*. These include several interferon-stimulated response elements sites for STAT1, IRF8, and IRF9. Our results also confirmed that age and gender play no significant role in the regulation of *ACE2* mRNA expression in the lung.

## Introduction

A zinc metalloenzyme, angiotensin-converting enzyme (ACE) was discovered 64 years ago and first named as a hypertension-converting enzyme [1]. Classically, ACE is well known for its roles in the regulation of arterial pressure through conversion of angiotensin I to active angiotensin II and cleavage of bradykinin and neurotensin [2]. As a zinc metalloenzyme, ACE belongs to a large cluster of zinc-binding proteins. The first zinc metalloenzyme, carbonic

**Data Availability Statement:** All relevant data are within the manuscript and its Supporting Information files.

**Funding:** The authors received no specific funding for this work.

**Competing interests:** The authors have declared that no competing interests exist.

anhydrase was discovered in 1932 by Meldrum and Roughton [3] and thereafter thousands of such metalloenzymes have been reported in different species of all phyla [4, 5].

Angiotensin-converting enzyme 2 (ACE2) was first discovered in 2000 when a novel homologue of ACE was cloned [2, 6, 7]. Although ACE and ACE2 share significant sequence similarity in their catalytic domains, they appear to act on different peptide substrates of angiotensins [8, 9]. Previous studies identified ACE2 as a functional receptor for severe acute respiratory syndrome corona virus 1 (SARS-CoV-1) which led to an outbreak of SARS infection in 2003 [10]. ACE2 is also a crucial receptor for the novel corona virus (SARS-CoV-2), which has caused a large global outbreak of COVID-19 infection with rapidly growing numbers of patients (32,968,853 confirmed cases as of September 28th, 2020, https://www.who.int/emergencies/diseases/novel-coronavirus-2019). A recent report suggested that soluble ACE2 fused to the Fc portion of immunoglobulin can neutralize SARS-CoV-2 *in vitro* [11]. This result was further confirmed by showing that human recombinant soluble ACE2 reduced SARS-CoV-2 infection on cultured Vero-E6 cells in a dose dependent manner [12]. Therefore, ACE2 also holds promise for treating patients with coronavirus infection.

The structural key for target cell infection by coronavirus is the viral spike (S) protein of SARS-CoV. ACE2 acts as a locking device for the virus, whereby the binding of the surface unit S1 facilitates viral attachment to the surface of target cells [13]. The cellular serine protease (TMPRSS2) promotes SARS-CoV entry via a dual mechanism. It cleaves both the SARS-CoV S protein and the virus receptor, ACE2, promoting both the viral uptake and the viral and cellular membrane fusion events [13–15]. The critical residues contributing to the receptor-spike protein interaction were first determined for SARS-CoV-1 [16] and recently in three independent studies for SARS-CoV-2 [17–19]. It has been proposed by biolayer interferometry studies that the receptor-binding domains of SARS-CoV-1 and SARS-CoV-2 S proteins bind with comparable affinities to human ACE2 [20]. In contrast, a modelling study suggested that binding of SARS-CoV-2 is stronger [21], which was convincingly confirmed by structural and biochemical data [17, 18].

The clinical characteristics of COVID-19 infection have recently been described based on data from 1,099 patients from mainland China [22]. It was found that the clinical characteristics of COVID-19 mimic those of SARS-CoV-1 infection. The most dominant symptoms include fever, cough, fatigue, and sputum production, whereas gastrointestinal symptoms are less common. In laboratory parameters, lymphopenia was detected in 83.2% of patients on admission. According to another recent survey of 278 patients with pneumonia caused by SARS-CoV-2, fever was the most common symptom, followed by cough [23]. Bilateral pneumonia has been detected by computed tomography scans in 67.0% of patients [24]. A recent study from Wuhan, China listed the most common clinical complications determined in critically ill COVID-19 patients [25]. The complications during clinical worsening included acute respiratory distress syndrome and respiratory failure, sepsis, acute cardiac injury, and heart failure.

Data on the localization of virus receptors can provide insight into mechanisms of virus entry, tissue tropism, and pathogenesis of the disease. Therefore, it is of particular interest to correlate COVID-19 symptoms with the distribution pattern of ACE2. The first studies performed by northern blotting indicated that ACE2 is located in the human heart, kidney, and testis [2]. Quantitative polymerase chain reaction (qPCR) showed the highest expression levels in the human cardiovascular system, testis, kidney, and intestine [26]. By immunohistochemistry, the expression of the ACE2 protein was identified in the human lung alveolar epithelial cells (type I and II pneumocytes), enterocytes of the small intestine, the brush border of the renal proximal tubules, the endothelial cells of arteries and veins, and arterial smooth muscle cells in several organs [27]. It was proposed that this distribution pattern of ACE2 could

explain the tissue tropism of SARS-CoV-1 for the lung, small intestine, and kidney [28]. On the other hand, the symptoms of COVID-19, in contrast to SARS-CoV-1 infection, are not associated to the same extent with the gastrointestinal tract in spite of the high expression of ACE2 in the intestinal enterocytes [29]. In COVID-19, diarrhea has been reported in just 3.8% of patients, in contrast to 40–70% in SARS-CoV-1 infection [22, 30]. A recent report indicated diarrhea in 18.1% of 254 COVID-19 patients [31].

There are conflicting reports on the expression of ACE2 in the upper respiratory tract [30]. Hamming and coworkers found that only the basal layer of nonkeratinized airway squamous epithelium shows positive signal [27], whereas Sims and colleagues demonstrated ACE2 expression on the luminal surface of ciliated cells in freshly excised human nasal and tracheo-bronchial tissue [32]. Ren and coworkers showed weak ACE2-positive signal in the epithelial cells of trachea and main bronchus [33]. Although lymphopenia is a typical feature of SARS [22, 30], ACE2 is not highly expressed on T or B cells or macrophages in the spleen or lymphoid organs [27].

It is known that both SARS-CoV and SARS-CoV-2 infections lead to worse outcome in the elderly [30, 34]. Recent studies have also indicated higher case fatality rates in males than females [35]. Therefore, one aim of the present study was to investigate whether age or gender could contribute to the regulation of ACE2 expression. We also decided to explore the transcriptional regulation of ACE2 gene expression using a novel computational tool recently developed by the first author of this article. Notably, data on ACE2 distribution is still conflicting, and thus we aimed to get a more comprehensive view of the cell types expressing the receptor of SARS-CoV-2. Finally, we studied the coexpression of ACE2 with other genes and explored its putative functions using a gene ontology enrichment analysis.

## Methods

### ACE2 mRNA expression

From the FANTOM5 project [36], cap analysis of gene expression (CAGE) sequencing of cDNA has been performed in 1,839 human samples from 875 different primary cells, tissues, and cell lines (description of all public datasets used presented as S1 Table). Expression of transcription start sites (TSSs) was extracted and combined for all genes in all samples as tags per million (TPM). From this compiled set, ACE2 gene expression was extracted and presented as barplot using the Matplotlib [37] and Seaborn [38] Python libraries. Similarly, human gene expression data (as TPM) was extracted from the GTEx database, which is an ongoing large-scale project to identify human variation, regulation, and gene expression [39], along with metadata on the samples. ACE2 gene expression values were separated by tissue and compared among 10-year interval age groups to determine if the values showed any differences throughout the lifecycle. Boxplots for tissues of relevance were generated using Matplotlib and Seaborn libraries.

### Coexpression and gene ontology enrichment analysis

In each of the tissues present in the GTEx dataset, expression values for ACE2 were compared with expression of all other genes by Spearman correlation analysis using the SciPy [40] Python library to identify those genes with concordant expression patterns. Bonferroni correction was used to derive an adjusted p-value threshold of 9.158E-07. For each tissue, those genes which both satisfied the Bonferroni-adjusted p-value threshold and had a correlation of expression of 0.50 or greater were analyzed using the Gprofiler gene ontology (GO) enrichment analysis [41] Python library to identify possible enriched terms in biological process

(BP), molecular function (MF), cellular component (CC), human phenotype (HP), KEGG pathway, and WikiPathways (WP) ontologies.

## ACE2 protein expression

Immunohistochemical localization of human ACE2 was evaluated from immunostained specimens provided by Protein Expression Atlas (https://www.proteinatlas.org/) [42]. The dataset included three specimens of duodenum, three specimens of kidney, three specimens of testis, three specimens of lung, and two specimens of nasopharynx. The images of the Fig 2 represent duodenum from 77-years-old female, kidney from 36-years-old male, testis from 38-years-old male, lung from 61-years-old female, and nasopharyngeal mucosa from 78-years-old female. According to Protein Expression Atlas the immunostainings were performed with the rabbit anti-human polyclonal antibody (HPA000288; Sigma Aldrich, St. Louis, MO) raised against 111 N-terminal amino acids of ACE2 and diluted 1:250 for the staining.

## Promoter analysis

Analysis of *ACE2* promoter regions was performed using the TFBSfootprinter tool (https://github.com/thirtysix/TFBS_footprinting) which uses transcription-relevant data from several major databases to enhance prediction of putative TFBSs, including: all cell types aggregated and merged human ATAC-Seq data from ENCODE [43], transcription start sites and expression data from FANTOM5 [44], expression quantitative trail loci from GTEx [39], TFBS metacluster data from GTRD [45], TFBS binding profile data from JASPAR [46], and sequence and conservation data from Ensembl [47]. Detailed description of this novel tool is under preparation [48]. Previous studies identified two distinct tissue-specific transcription start sites (TSS) for intestine and lung expression [49], which correspond to primary protein-coding Ensembl transcripts ENST00000252519 and ENST00000427411, respectively. These two transcripts were targeted for transcription factor binding site (TFBS) analysis; first with a scan for all 575 Jaspar TFs and input parameters of 1,000 base pairs (bp) upstream and 200 bp downstream (relative to the TSS); secondly with a limited set of 15 interferon-stimulated TF genes and a broader area of 1,500 bp upstream and 500 bp downstream. Likewise, an analysis of the promoter region of a putative new short form of *ACE2* was performed.

## Single-cell RNA-Seq

Single-cell expression datasets were identified for relevant tissues/cells of lung (human) [50], trachea (mouse) [51], and small intestine (mouse) [52]. Using a modified workflow described previously in [53], for each dataset the samples were filtered by Gaussian fit of read count, expressed gene count, and number of cells in which a gene is expressed. Counts were normalized by cell, log transformed, principle component analysis performed with 15 components, and k-nearest neighbors computed using SCANPY [54], and then the full dataset normalized with R package 'scran' [55]. Batch correction by individual and sample region was performed with SCANPY using the ComBat function. The top 1,000 genes with highly differential expression were identified for cluster analysis which was performed with Uniform Manifold Approximation and Projection (UMAP) and force directed graph models. The top 100 marker genes were identified as those with higher expression unique to each cluster by Welch t-test in SCANPY. Expression of the *ACE2* gene was mapped onto cluster figures to determine overlap with previously identified cell types or cell type marker genes identified in the literature. Cell type was mapped by expression of known marker genes of cell types expressed in the lung and small intestine, as defined by *de novo* prediction in the original articles.

## Statistics

Comparisons of *ACE2* expression values in different tissues and between groups delineated by age or sex, were carried out by one-way ANOVA using the stats package in the SciPy [40] Python library. Only groups with 20 or more observations and a 2-sided chi squared probability of normality of $< = 0.1$ (due to the robustness of ANOVA to non-normal distributions) were used for comparison. Correlation of gene expression values was calculated by two-sided Spearman rank-order analysis, where a Bonferroni-corrected p-value threshold was computed using $\alpha = 0.05/$ number of comparisons. Gene ontology enrichment analyses performed using the GProfiler tool utilize a custom algorithm for multiple testing of dependent results, which corresponds to an experiment-wide threshold of $\alpha = 0.05$. TFBSfootprinter analysis of the *ACE2* promoter limits results for individual TFBSs whose score satisfies a genome-wide threshold of $\alpha = 0.01$.

## Results

### ACE2 is weakly expressed in the lung

The first aim of our study was to investigate different human tissues using publicly available datasets for the distribution of *ACE2* mRNA and protein. In the FANTOM5 dataset, the highest values for *ACE2* mRNA, ranked according to signal intensity, were seen for the small intestine, dura mater, colon, testis, thalamus, and rectum (Fig 1).

Fig 2 shows the expression of ACE2 protein in selected human tissues. Representative example images of the ACE2 immunostaining were prepared from tissue specimens of the Human Protein Atlas database (https://www.proteinatlas.org/). The results indicate a strong signal for ACE2 protein in the brush border of small intestinal enterocytes. In the kidney, strong immunostaining reactions were present in the epithelial cells of proximal convoluted tubules and Bowman´s capsule. The seminiferous tubules and interstitial cells of testis also demonstrated strong immunostaining. No immunoreactions for ACE2 were observed in the lung specimens. Very weak signal, associated with apical membranes, was detected in sporadic ciliary cells of a nasopharyngeal mucosa sample. Although the evaluation of immunostaining reaction is generally considered semiquantitative at most, the results seem to correlate fairly well with the corresponding mRNA expression levels.

### Single cell RNA-Seq analysis indicates cell-specific expression for *ACE2* mRNA

The respiratory tract is the main target region that is affected by COVID-19 infection. Bulk RNA-Seq data from lung specimens showed low expression levels for *ACE2* (Fig 1). Therefore, we performed an analysis of single cell RNA-Seq using both human lung and mouse trachea datasets, representing the breadth of the lower respiratory tract. Figs 3 and 4 show the expression of *ACE2* mRNA in identified cell types of lung and trachea, respectively. In lung, *ACE2* expressing cells are generally uncommon with no cell type having a ratio of *ACE2*-expressing cells greater than 2.6%. The cell types with the greatest proportion of *ACE2* expression are those of arterial vascular endothelial cells (2.55%), goblet cells (2.02%), and venous vascular endothelial cells (1.33%). In trachea, the highest ratio of *ACE2*-expressing cells included the club cells (16.62%), goblet cells (13.84%), and ciliary epithelial cells (6.63%).

Since both the airways and intestine contain goblet cells, SARS-CoV-1 affects gastrointestinal tract, and bulk RNA-Seq data shows high expression in small intestine and colon, we decided to analyze another single cell RNA-Seq dataset covering mouse intestinal epithelial cells. Fig 5 indicates the highest levels of *ACE2* mRNA signal in the absorptive enterocytes (44.09%), whereas the intestinal goblet cells (1.35%) remain mostly negative.

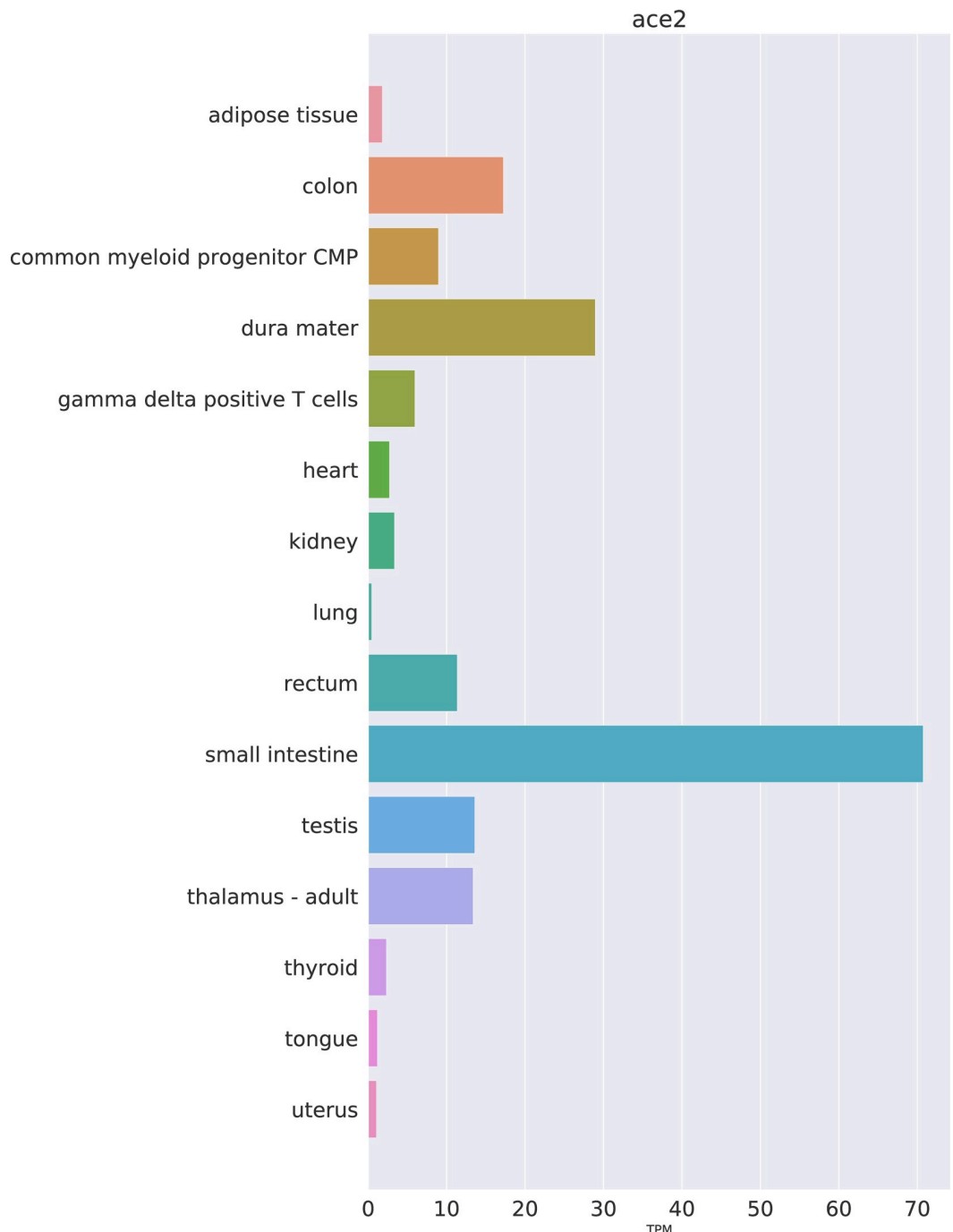

**Fig 1. Expression of *ACE2* mRNA in selected human tissues.** Expression values as TPM have been extracted from the FANTOM5 dataset.

### *ACE2* mRNA expression levels are unrelated to age and gender in the lung

Since both age and gender may contribute to onset and severity of COVID-19 symptoms we aimed to investigate the effect of these variables on the expression levels of *ACE2* mRNA. Fig 6 indicates that some tissues showed a slight trend to lower expression in older age categories. Among all tested tissues, statistically significant differences between the age categories were

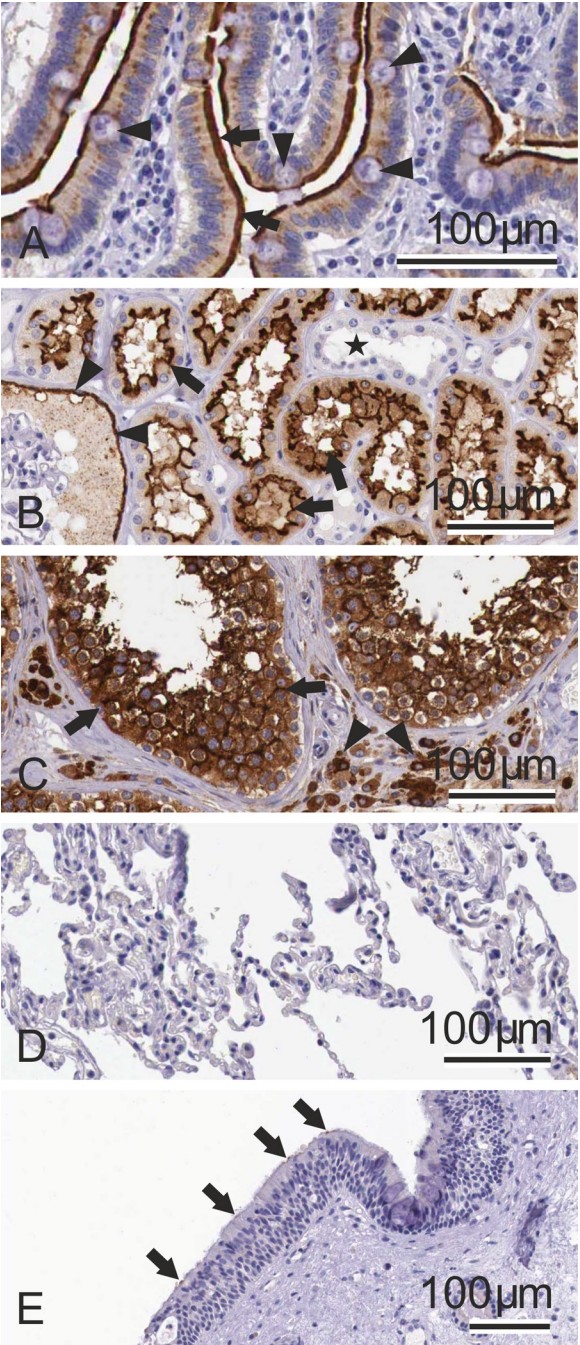

**Fig 2. Immunohistochemical localization of ACE2 protein in selected human tissues.** In the duodenum (A), the protein is most strongly localized to the apical plasma membrane of absorptive enterocytes (arrows). The goblet cells (arrowheads) show weaker apical staining. Intracellular staining is confined to the absorptive enterocytes. In the kidney (B), ACE2 shows strong apical staining in the epithelial cells of the proximal convoluted tubules (arrows) and Bowman´s capsule epithelium (arrowheads). The distal convoluted tubules are negative (asterisk). The testis specimen (C) shows strong immunostaining in the seminiferous tubules (arrows) and interstitial cells (arrowheads). The lung sample (D) is negative. In the nasopharyngeal mucosa (E), ACE2 signal is very weak and only occasional epithelial cells show weak signals (arrows). Immunostained specimens were taken from the Protein Expression Atlas (https://www.proteinatlas.org/).

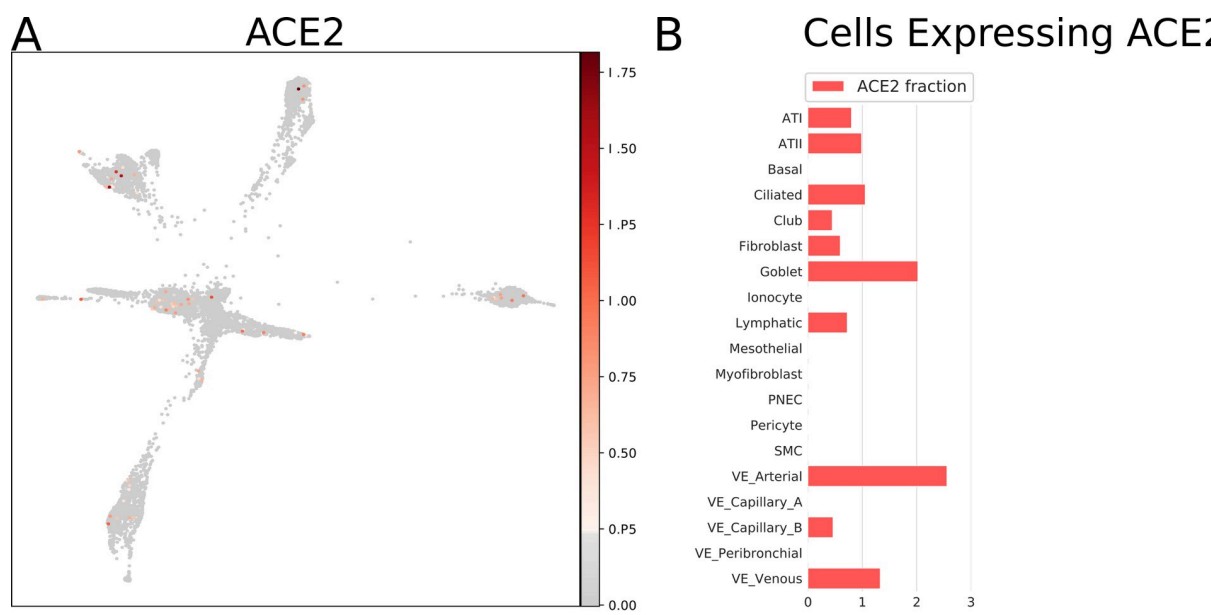

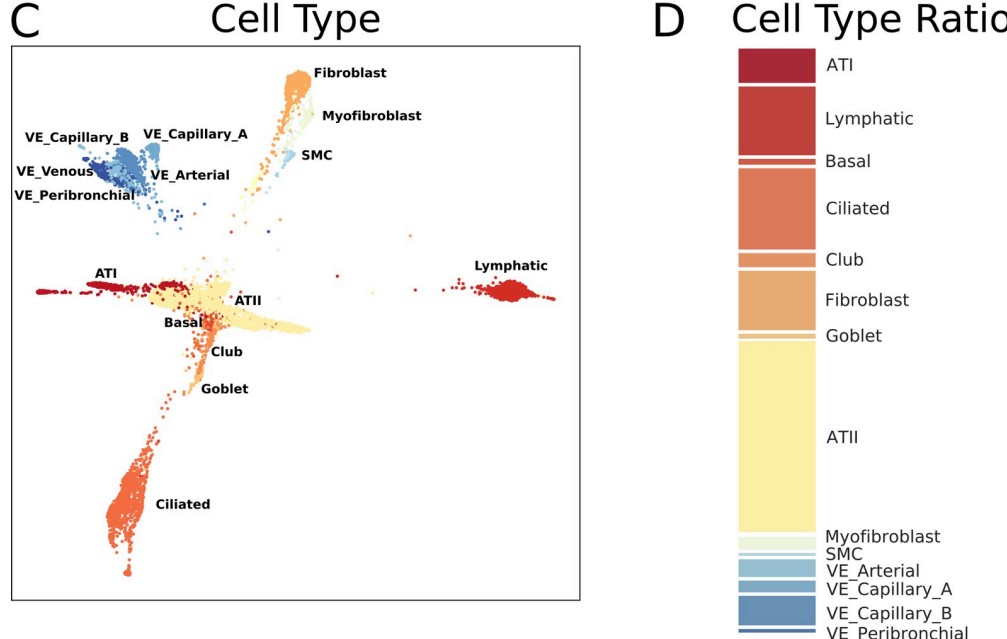

**Fig 3. Single cell RNA-Seq analysis of different cell types from the lung (human), derived from data from GEO dataset GSE136831 [50].**
*ACE2* mRNA expression as normalized, batch-corrected counts is shown for comparison in upper panel. The force directed layout plot was computed and visualized in ScanPy [54]. For each cell type the ratio of cells expressing *ACE2* is presented in addition to a stacked barplot of the relative cell type frequencies in the whole dataset. Alveolar type I (ATI), alveolar type II (ATII), pulmonary neuroendocrine cells (PNEC), smooth muscle cells (SMC), vascular endothelia (VE).

seen in the tibial nerve (p = 8.58 x $10^{-6}$), minor salivary gland (p = 0.002), aorta (p = 0.003), whole blood (p = 0.005), transverse colon (p = 0.010), hypothalamus (p = 0.039), and sun exposed skin (p = 0.046). Importantly, the lung specimens showed no significant difference of

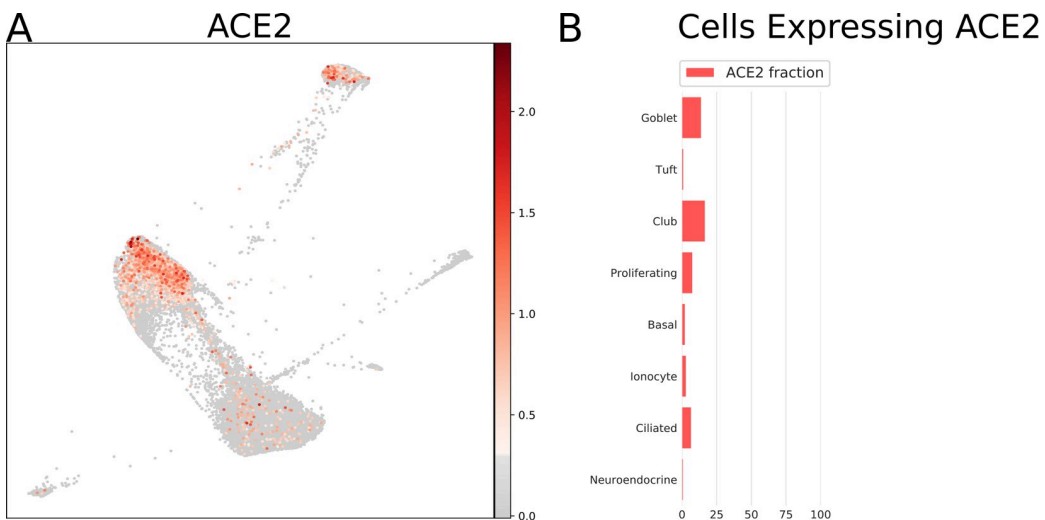

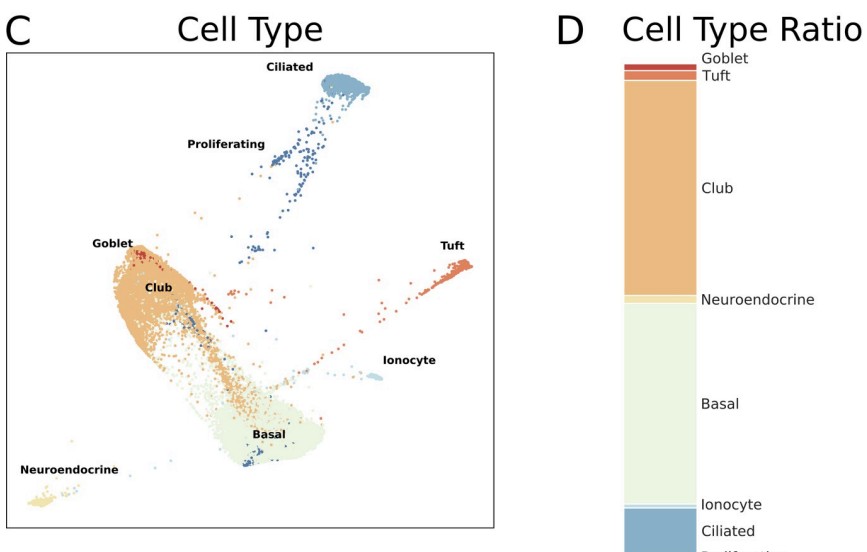

**Fig 4. Single cell RNA-Seq analysis of different cell types from the respiratory tract (mouse tracheal epithelium), derived from data from GEO dataset GSE103354 [51].** *ACE2* mRNA expression as normalized, batch-corrected counts is shown for comparison in upper panel. The force directed layout plot was computed and visualized in ScanPy [54]. For each cell type the ratio of cells expressing *ACE2* is presented in addition to a stacked barplot of the relative cell type frequencies in the whole dataset.

*ACE2* mRNA expression between different age categories (p = 0.681). Complete data on *ACE2* mRNA expression levels in different age categories are shown in S2 Table. To make a binary comparison of expression by age, samples were divided into groups of <50 and ≥50 years of age. In comparison of these younger and older age groups, significant differences in expression were found in tibial nerve (p = $2.47 \times 10^{-7}$), whole blood (p = $3.21 \times 10^{-4}$), minor salivary gland (p = $4.89 \times 10^{-4}$), sun exposed skin (p = 0.003), transverse colon (p = 0.022), testis (p = 0.025), esophageal muscle layer (p = 0.040), and subcutaneous adipose tissue (p = 0.045). Additionally, with the same age groups, comparisons were made for both males and females (S3 Table). In males, *ACE2* expression was lower in the ≥50 age group for tibial nerve

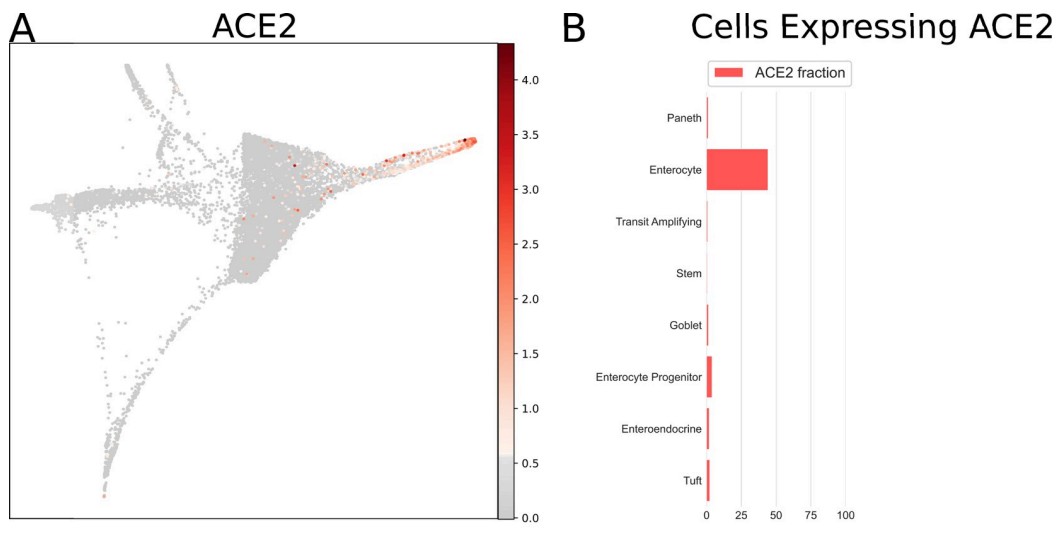

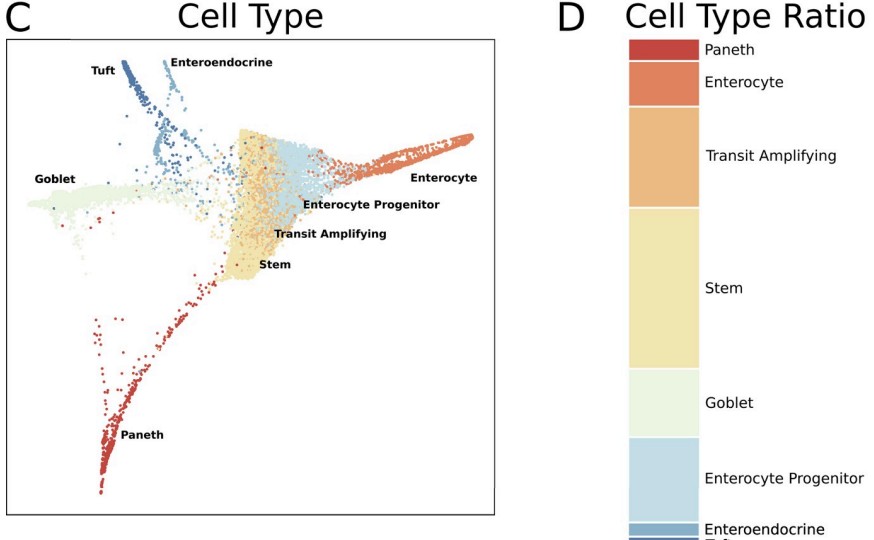

**Fig 5. Single cell RNA-Seq analysis of mouse intestinal epithelial cells.** Data is from GEO dataset GSE92332 [52]. *ACE2* mRNA expression as normalized, batch-corrected counts is shown for comparison in upper panel. The force directed layout plot was computed and visualized in ScanPy [54]. For each cell type the ratio of cells expressing *ACE2* is presented in addition to a stacked barplot of the relative cell type frequencies in the whole dataset.

(p = 1.33 x $10^{-7}$), subcutaneous adipose tissue (p = 1.23 x $10^{-4}$), minor salivary gland (p = 4.03 x $10^{-4}$), transverse colon (p = 4.67 x $10^{-3}$), sigmoid colon (p = 0.017), testis (p = 0.025), visceral adipose tissue of omentum (p = 0.026), sun exposed skin (p = 0.027), whole blood (p = 0.032), and bladder (p = 0.042); while increased in coronary artery (p = 0.015). In females, *ACE2* expression was lower in the ≥50 age group in whole blood (p = 0.005) and sun exposed skin (p = 0.049); and higher in esophagus (p = 0.007) and terminal ileum (p = 0.022).

　　*ACE2* mRNA levels largely overlapped between male and female sexes as shown in Fig 7. In the lung, no statistically significant difference was observed in the expression levels between the male and female subjects (p = 0.908). Statistically significant differences were observed in the adipose tissue (p = 0.0001), whole blood (p = 0.0002), amygdala (p = 0.0006), transverse colon (p = 0.0008),

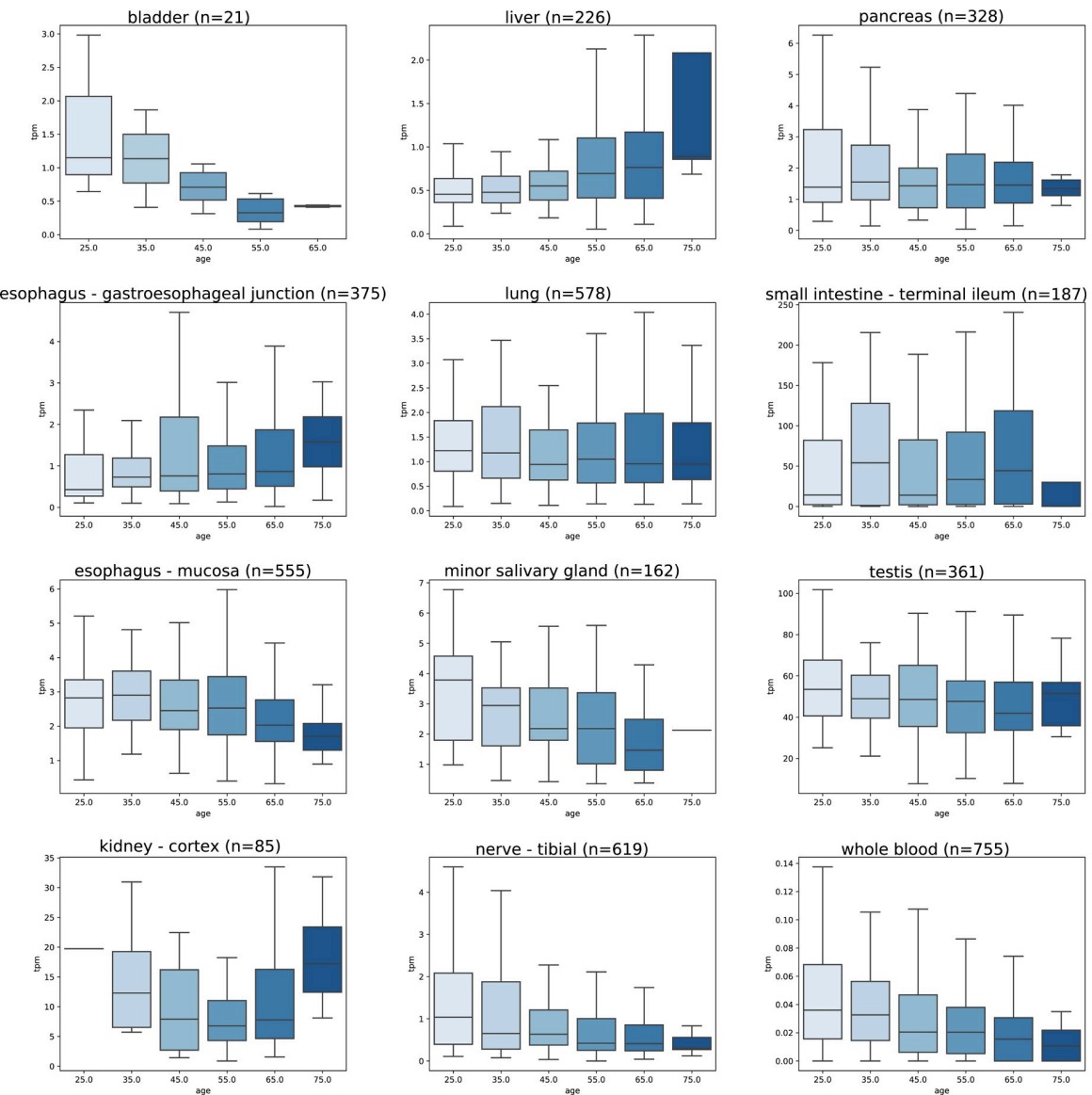

**Fig 6. Effect of age on *ACE2* mRNA expression levels.** Data is extracted from the GTEx dataset as TPM. In these organs, ANOVA revealed significant differences between age categories in tibial nerve (p = 8.58 x 10$^{-6}$), minor salivary gland (p = 0.002), and whole blood (p = 0.005). In other tissues, the differences did not reach statistical significance. The highest TPM values are seen in the small intestine, testis, and kidney.

muscle layer of esophagus (p = 0.002), left ventricle of heart (p = 0.005), Epstein-Barr virus-transformed lymphocytes (p = 0.015), and esophagus-gastroesophageal junction (p = 0.024). Notably, there was no clear sex-specific trend pointing to one direction in all these cases. *ACE2* mRNA expression levels in all studied tissues sorted according to subjects´ gender are shown in S4 Table.

## Proximal promoter contains putative TFBSs for ileum, colon, and kidney expression

TFBS analysis of the *ACE2* intestinal transcript promoter (ENST00000252519) revealed several candidate binding sites which occur in a cluster extending from 400 bp upstream of the

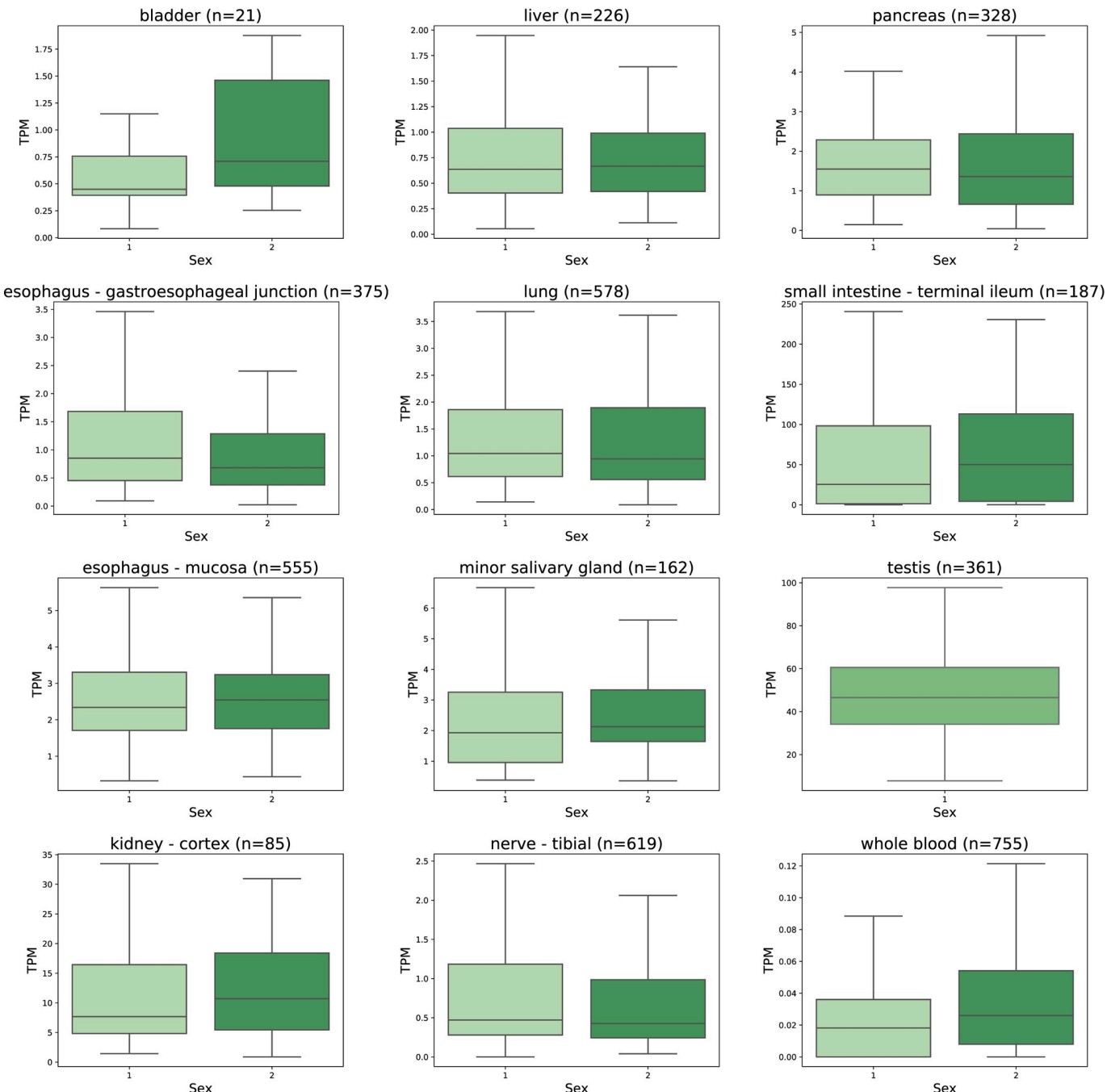

**Fig 7. Effect of gender on *ACE2* mRNA expression levels.** Data is extracted from the GTEx dataset as TPM. The expression levels in males and females overlap in all tissue categories. Statistically significant differences studied by ANOVA analysis were determined in esophagus-gastroesophageal junction (p = 0.024) and whole blood (p = 0.0002). The *ACE2* mRNA expression levels in testis specimens are shown here for comparison. 1 = male, 2 = female.

transcription start site; CDX2, HNF1A, FOXA1, SOX4, TP63, HNF4A, DUX4, FOXA2, NR2F6, and SOX11 (Fig 8A). These predicted sites overlap an evolutionarily conserved region in mammals and are proximal to several ATAC-Seq peaks. In several tissues these TFs are found to be highly positively correlated (>0.7) with expression of *ACE2*, as determined using RNA-Seq gene expression values from the GTEX dataset: *CDX2* (colon, terminal ileum),

*HNF1A* (colon, kidney, terminal ileum), *FOXA1* (cervix, colon, terminal ileum), *HNF4A* (colon, terminal ileum), *FOXA2* (colon, kidney), *NR2F6* (colon, kidney, terminal ileum), and *SOX11* (kidney). In addition, two of the TFs are highly negatively correlated with *ACE2* expression *DUX4* (kidney) and *FOXA1* (kidney). Full prediction results are included in S5 Table and TF correlations by tissue are present in S6 Table. An expanded analysis for binding sites of interferon-stimulation mediating TF genes in the region 1,500 bp upstream to 500 bp downstream of the *ACE2* intestinal transcript's TSS revealed putative binding sites for STAT1 (3x), STAT3 (4x), STAT5A:STAT5B dimer (3x), STAT4, STAT1:STAT2 dimer (2x), IRF2, IRF3, IRF4 (2x), IRF5 (2x), IRF8 (3x), and IRF9 (S1 Fig). Each of these predictions satisfied a PWM p-value threshold of <0.0001, while only the STAT1, IRF9, and IRF8 sites also satisfied a combined affinity score p-value of <0.05.

Analysis of the *ACE2* lung transcript promoter (ENST00000427411) produced putative TFBS predictions for ESRRA, HNF4A, CDX2, CEBPA, ESRRB, MEF2B, TCF7, TCF7L2, JUN, and LEF1 (Fig 8B). Full prediction results are included in S5 Table. The predicted TFBSs clustered within 200 base pairs of the TSS, and overlap with evolutionarily conserved regions, TFBS metaclusters, and ATAC-Seq peaks. The TFs corresponding to predicted TFBSs, which are positively correlated (>0.7) with *ACE2* expression, are *ESRRA* (terminal ileum, colon), *HNF4A* (terminal ileum, colon), *CDX2* (colon, terminal ileum), *CEBPA* (colon, terminal ileum), *ESRRB* (cervix), *TCF7L2* (testis). Those TFBSs with TFs which strongly (<-0.7) negatively correlate with *ACE2* are *ESRRA* (kidney) and *TCFL72* (kidney).

The lung-specific transcript TSS aligns with the p3@*ACE2* FANTOM5 dataset CAGE peak, which indicates that the expression of this transcript is much lower than the intestinal transcript, which corresponds with p1@*ACE2* and p2@*ACE2* FANTOM5 CAGE peaks. Common between the two tissue-specific transcripts, are predictions for CDX2 and HNF-family transcription factors. An expanded analysis for binding sites of interferon-stimulation mediating TF genes in the region 1,500 bp upstream to 500 bp downstream of the lung transcript's TSS revealed putative binding sites for STAT1:STAT2 dimer, STAT1 (2x), STAT3, STAT4, STAT6 (2x), and IRF8 (S1 Fig). Each of these predictions satisfied a PWM p-value threshold of <0.0001, while only the IRF8 site also satisfied a combined affinity score p-value of <0.05.

TFBS analysis of the recently identified putative short form *ACE2* transcript [56–58], with a TSS between exons 9 and 10 of the canonical gene, produced predictions for IRF9, IRF8, JUND, FOSL1, GATA1, JUNB, IRF4, JUN, and FOS, among others (S5 Table). Within the first -56 to -31 bp upstream of the short-form *ACE2* TSS, are overlapping binding sites for several IRF TFs and a STAT1:STAT2 dimer, while further upstream are binding sites for several STAT TFs at -662 to -647 bp and -911 to –897 (Fig 9). Each of these predictions satisfied a PWM p-value threshold of <0.001, while only the IRF9, IRF8, and IRF4 sites also satisfied a combined affinity score p-value of <0.05.

## ACE2 mRNA expression correlates with metalloproteases and transporter genes

Coexpression analysis identified numerous genes in ileum, testis, colon, and kidney which are highly correlated (>0.8) with *ACE2* (Tables 1 and S7). In particular, in the ileum there are a number of genes with correlation values greater than 0.95. In contrast, analysis of the lung shows a maximum correlation of expression of 0.6275. The genes with which *ACE2* mRNA expression shows the highest levels of coexpression code for metalloprotease and transporter proteins. Selected tissue-specific *ACE2*-correlated genes, determined with bulk RNA-Seq data, are presented with their expression levels within the scRNA-Seq trachea and intestinal datasets in Fig 10.

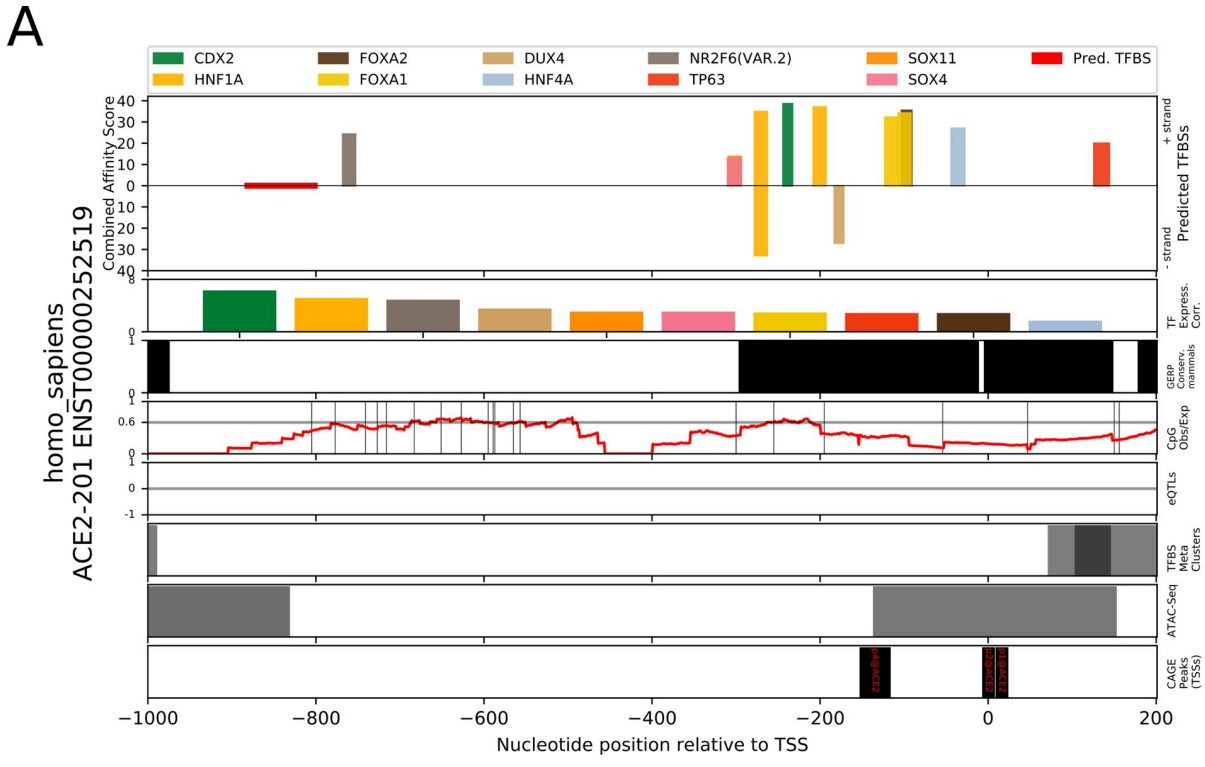

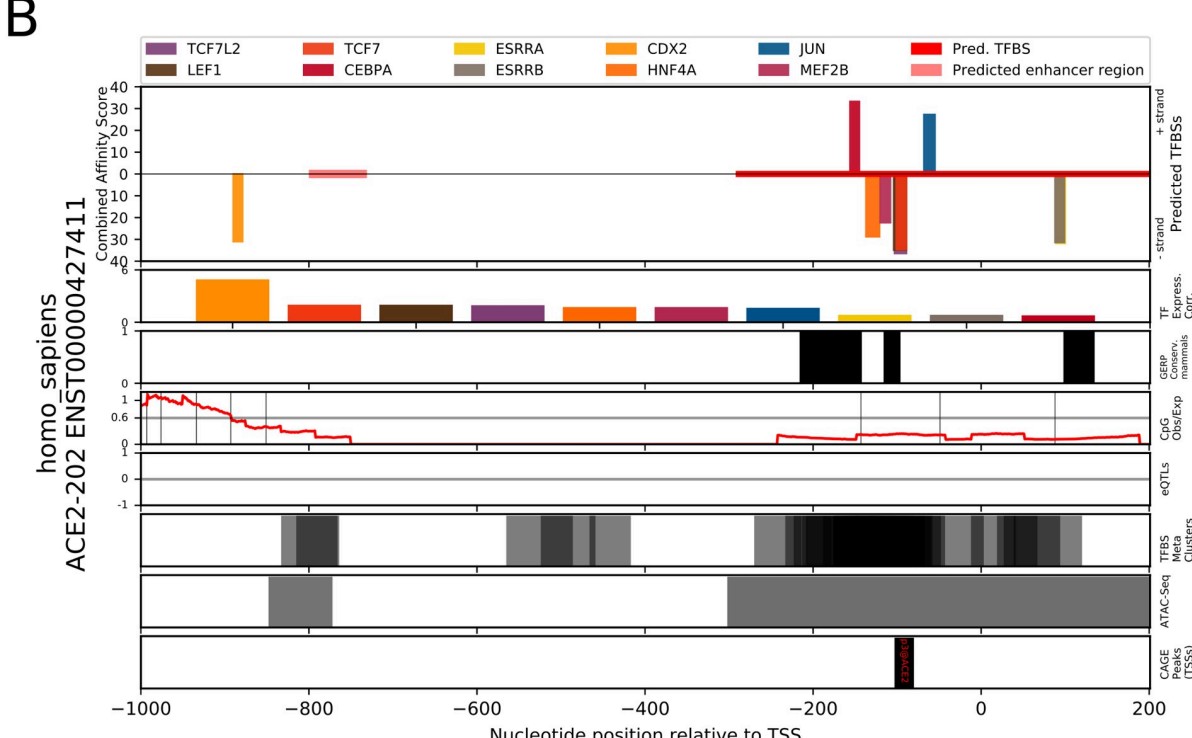

**Fig 8. Prediction of transcription factor binding sites in the human *ACE2* gene promoter regions of Ensembl transcripts ENST00000252519 and ENST00000427411 using TFBSfootprinter.** A) Promoter region of the intestine specific *ACE2* transcript. The results show putative binding sites for several transcription factors which have a strong correlation of expression with *ACE2* in colon, kidney, and ileum. The predicted binding sites overlap regions of conservation in mammal species (Ensembl GERP), and cluster within 400 base pairs (bp) of the transcription start site. B) Promoter region of the lung specific *ACE2* transcript. The predicted binding sites cluster within 200 bp of the TSS and overlap regions of conservation in mammal species, ATAC-Seq peaks (ENCODE), and TFBS metaclusters (GTRD).

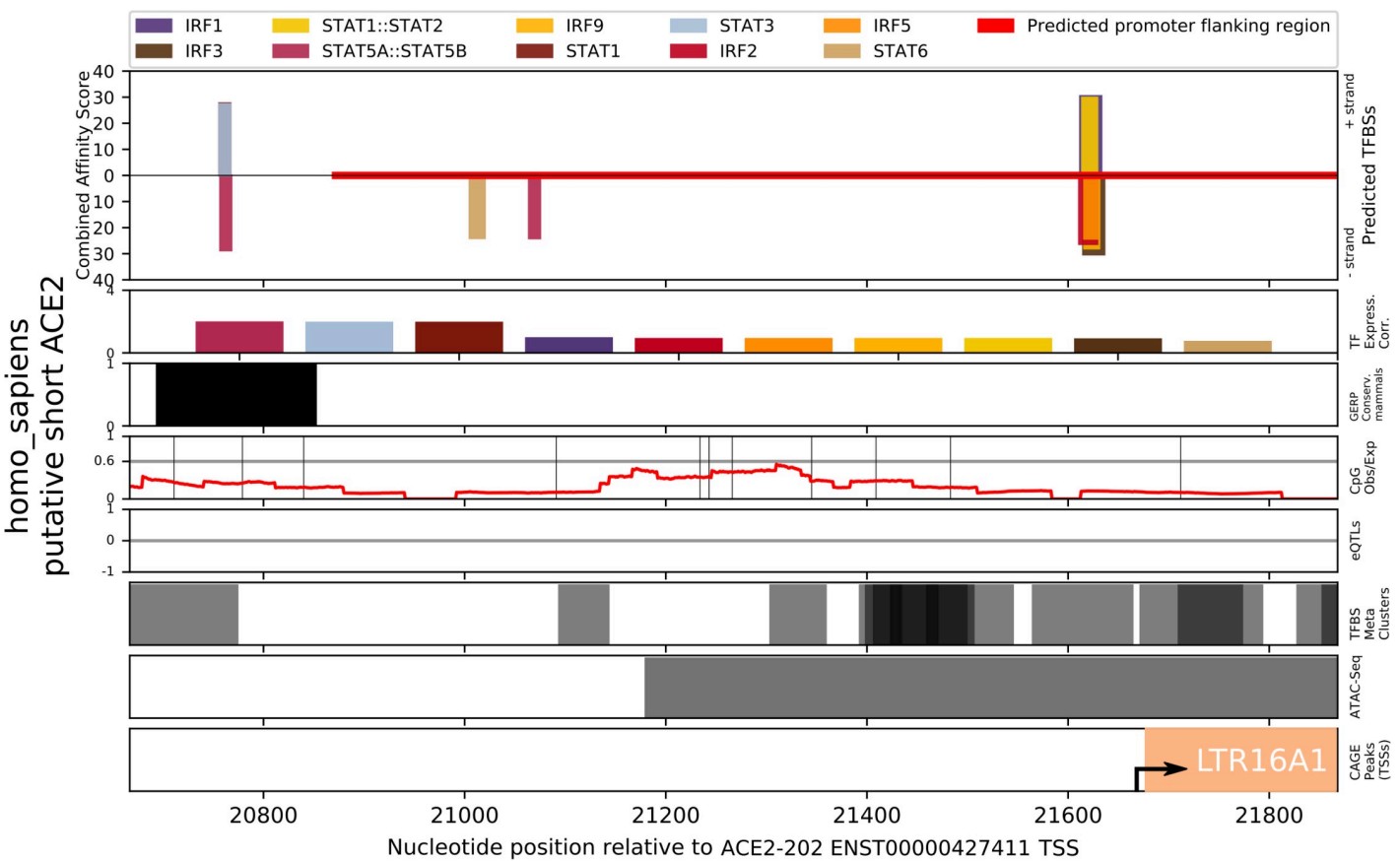

**Fig 9. Prediction of interferon-stimulation mediating transcription factor binding sites in the promoter region of the putative short-form of human *ACE2* gene using TFBSfootprinter.** The long terminal repeat 16A1 (LTR16A1) is cataloged in the FANTOM dataset [36] and has been identified as potentially relevant for interferon-mediated transcription of this new *ACE2* transcript [56]; it is displayed in the CAGE peaks track along with an arrow indicating the TSS. The region analyzed represents –1,000 bp to +200 bp relative to the putative TSS, while nucleotide positions at bottom are given relative to the *ACE2* full length transcript.

## ACE2 is associated with vascular growth

GO enrichment analysis of *ACE2* mRNA expression in all tissues produced 22 terms which were enriched in BP, CC, HP, KEGG, and WP ontologies (Table 2). A total of 12 of these terms were related to blood vessel growth, including the three most strongly enriched terms, 'angiogenesis [GO:0001525]', 'blood vessel morphogenesis [GO:0048514]', and 'vasculature development [GO:0072358]'. Full GO enrichment results for relevant tissues (lung, small intestine, kidney, colon, and testis) are included as S8 Table.

## Discussion

The predominant pathological features of COVID-19 infection largely mimic those previously reported for SARS-CoV-1 infection. They include dry cough, persistent fever, progressive dyspnea, and in some cases acute exacerbation of lung function with bilateral pneumonia [32]. Major lung lesions include several pathological signs, such as diffuse alveolar damage, inflammatory exudation in the alveoli and interstitial tissue, hyperplasia of fibrous tissue, and eventually lung fibrosis [59–61]. It has been shown by fluorescence *in situ* hybridization technique that SARS-CoV-1 RNA locates to the alveolar pneumocytes and alveolar space [62, 63]. Mossel and colleagues demonstrated that SARS-CoV-1 replicates in type 2 (AT2) pneumocytes, but

**Table 1. Genes associated with *ACE2* mRNA expression in selected human tissues.** Derived from GTEx bulk RNA-Seq data [39].

| Tissue | Correlated_gene | Correlation | p-value | HGNC | UniProt | Description | Panther protein class |
|---|---|---|---|---|---|---|---|
| terminal ileum | *ENPEP* | 0.9653 | 8.29E-110 | 3355 | Q07075 | Glutamyl aminopeptidase | metalloprotease (PC00153) |
| terminal ileum | *MEP1B* | 0.9653 | 8.35E-110 | 7020 | Q16820 | Meprin A subunit beta | metalloprotease (PC00153) |
| terminal ileum | *SLC15A1* | 0.9621 | 2.19E-106 | 10920 | P46059 | Solute carrier family 15 member 1 | transporter (PC00227) |
| terminal ileum | *MEP1A* | 0.9619 | 4.4E-106 | 7015 | Q16819 | Meprin A subunit alpha | metalloprotease (PC00153) |
| terminal ileum | *SLC3A1* | 0.9608 | 5.63E-105 | 11025 | Q07837 | Neutral and basic amino acid transport protein rBAT | amylase (PC00048) |
| terminal ileum | *APOB* | 0.9587 | 6.45E-103 | 603 | P04114 | Apolipoprotein B-100 | |
| terminal ileum | *SLC6A19* | 0.9581 | 2.33E-102 | 27960 | Q695T7 | Sodium-dependent neutral amino acid transporter B(0)AT1 | primary active transporter (PC00068) |
| terminal ileum | *ANPEP* | 0.9562 | 1.16E-100 | 500 | P15144 | Aminopeptidase N | metalloprotease (PC00153) |
| terminal ileum | *MTTP* | 0.9559 | 2.45E-100 | 7467 | P55157 | Microsomal triglyceride transfer protein large subunit | transporter (PC00227) |
| terminal ileum | *TUBAL3* | 0.9554 | 6.24E-100 | 23534 | A6NHL2 | Tubulin alpha chain-like 3 | tubulin (PC00228) |
| testis | *PLP1* | 0.9092 | 1.07E-138 | 9086 | P60201 | Myelin proteolipid protein | myelin protein (PC00161) |
| colon-transverse | *MEP1A* | 0.9056 | 1.52E-152 | 7015 | Q16819 | Meprin A subunit alpha | metalloprotease (PC00153) |
| kidney | *TINAG* | 0.9012 | 7.07E-32 | 14599 | Q9UJW2 | Tubulointerstitial nephritis antigen | cysteine protease (PC00081) |
| colon-transverse | *GDA* | 0.8964 | 8.19E-145 | 4212 | Q9Y2T3 | Guanine deaminase | deaminase (PC00088) |
| colon-transverse | *MGAM2* | 0.8963 | 9E-145 | 28101 | Q2M2H8 | Probable maltase-glucoamylase 2 | glucosidase (PC00108) |
| colon-transverse | *TMEM236* | 0.894 | 6.66E-143 | 23473 | Q5W0B7 | Transmembrane protein 236 | |
| colon-transverse | *GDPD2* | 0.8935 | 1.58E-142 | 25974 | Q9HCC8 | Glycerophosphoinositol inositolphosphodiesterase GDPD2 | |
| colon-transverse | *PLS1* | 0.8932 | 2.81E-142 | 9090 | Q14651 | Plastin-1 | non-motor actin binding protein (PC00165) |
| colon-transverse | *HHLA2* | 0.893 | 3.59E-142 | 4905 | Q9UM44 | HERV-H LTR-associating protein 2 | immunoglobulin receptor superfamily (PC00124) |
| colon-transverse | *SLC3A1* | 0.8928 | 5.27E-142 | 11025 | Q07837 | Neutral and basic amino acid transport protein rBAT | amylase (PC00048) |
| colon-transverse | *COL17A1* | 0.8914 | 6.12E-141 | 2194 | Q9UMD9 | Collagen alpha-1(XVII) chain | extracellular matrix structural protein (PC00103) |
| testis | *NDFIP1* | 0.8911 | 2.91E-125 | 17592 | Q9BT67 | NEDD4 family-interacting protein 1 | |
| testis | *SPARC* | 0.8908 | 4.67E-125 | 11219 | P09486 | SPARC | extracellular matrix glycoprotein (PC00100) |
| colon-transverse | *TINAG* | 0.8905 | 2.94E-140 | 14599 | Q9UJW2 | Tubulointerstitial nephritis antigen | cysteine protease (PC00081) |
| testis | *SCP2* | 0.8886 | 1.34E-123 | 10606 | P22307 | Non-specific lipid-transfer protein | transfer/carrier protein (PC00219) |
| testis | *SMARCA1* | 0.8852 | 2.44E-121 | 11097 | P28370 | Probable global transcription activator SNF2L1 | DNA helicase (PC00011) |
| testis | *ATP2B1* | 0.8829 | 6.45E-120 | 814 | P20020 | Plasma membrane calcium-transporting ATPase 1 | primary active transporter (PC00068) |

*(Continued)*

**Table 1.** (Continued)

| Tissue | Correlated_gene | Correlation | p-value | HGNC | UniProt | Description | Panther protein class |
|---|---|---|---|---|---|---|---|
| **testis** | VSIG1 | 0.8815 | 4.6E-119 | 28675 | Q86XK7 | V-set and immunoglobulin domain-containing protein 1 | |
| **testis** | HTATSF1 | 0.8809 | 1.09E-118 | 5276 | O43719 | HIV Tat-specific factor 1 | RNA splicing factor (PC00148) |
| **testis** | ACO1 | 0.8808 | 1.26E-118 | 117 | P21399 | Cytoplasmic aconitate hydratase | RNA binding protein (PC00031) |
| **testis** | ENPP5 | 0.8802 | 3.01E-118 | 13717 | Q9UJA9 | Ectonucleotide pyrophosphatase/phosphodiesterase family member 5 | nucleotide phosphatase (PC00173) |
| **kidney** | METTL7B | 0.8705 | 2.78E-27 | 28276 | Q6UX53 | Methyltransferase-like protein 7B | methyltransferase (PC00155) |
| **kidney** | ANPEP | 0.8683 | 5.3E-27 | 500 | P15144 | Aminopeptidase N | metalloprotease (PC00153) |
| **kidney** | LRP2 | 0.8671 | 7.59E-27 | 6694 | P98164 | Low-density lipoprotein receptor-related protein 2 | |
| **kidney** | SLC13A1 | 0.8667 | 8.5E-27 | 10916 | Q9BZW2 | Solute carrier family 13 member 1 | secondary carrier transporter (PC00258) |
| **kidney** | UGT3A1 | 0.8608 | 4.52E-26 | 26625 | Q6NUS8 | UDP-glucuronosyltransferase 3A1 | |
| **kidney** | SLC27A2 | 0.8535 | 3.26E-25 | 10996 | O14975 | Very long-chain acyl-CoA synthetase | secondary carrier transporter (PC00258) |
| **kidney** | TMEM27 | 0.8526 | 4.1E-25 | 29437 | Q9HBJ8 | Collectrin | |
| **kidney** | CLRN3 | 0.8443 | 3.36E-24 | 20795 | Q8NCR9 | Clarin-3 | |
| **kidney** | ACP5 | 0.8367 | 2.05E-23 | 124 | P13686 | Tartrate-resistant acid phosphatase type 5 | |
| **lung** | PARM1 | 0.6275 | 1.32E-64 | 24536 | Q6UWI2 | Prostate androgen-regulated mucin-like protein 1 | |
| **lung** | CNR1 | 0.6055 | 4.21E-59 | 2159 | P21554 | Cannabinoid receptor 1 | G-protein coupled receptor (PC00021) |
| **lung** | NUDT16 | 0.5918 | 6.52E-56 | 26442 | Q96DE0 | U8 snoRNA-decapping enzyme | |
| **lung** | DUOX1 | 0.5817 | 1.23E-53 | 3062 | Q9NRD9 | Dual oxidase 1 | oxidase (PC00175) |
| **lung** | DUOXA1 | 0.5739 | 6.2E-52 | 26507 | Q1HG43 | Dual oxidase maturation factor 1 | |
| **lung** | FMN1 | 0.571 | 2.53E-51 | 3768 | Q68DA7 | Formin-1 | |
| **lung** | TMEM164 | 0.5692 | 6.05E-51 | 26217 | Q5U3C3 | Transmembrane protein 164 | |
| **lung** | HSD17B4 | 0.5673 | 1.54E-50 | 5213 | P51659 | Peroxisomal multifunctional enzyme type 2 | |
| **lung** | MGST1 | 0.5631 | 1.16E-49 | 7061 | P10620 | Microsomal glutathione S-transferase 1 | |
| **lung** | SHMT1 | 0.5594 | 6.77E-49 | 10850 | P34896 | Serine hydroxymethyltransferase, cytosolic | methyltransferase (PC00155) |

not in type 1 (AT1) cells [64]. Considering all of these facts, it is not surprising that most histopathological analyses have been focused on distal parts of the respiratory airways, while the regions other than the alveolus have been less systematically studied.

To understand better the pathogenesis of COVID-19 we need to know where ACE2, the receptor for SARS-CoV, is located within the human respiratory tract and elsewhere. Overall, different studies including ours have convincingly shown that several organs, such as the small intestine, colon, kidney, and testis, express higher levels of *ACE2* than the lung and other parts

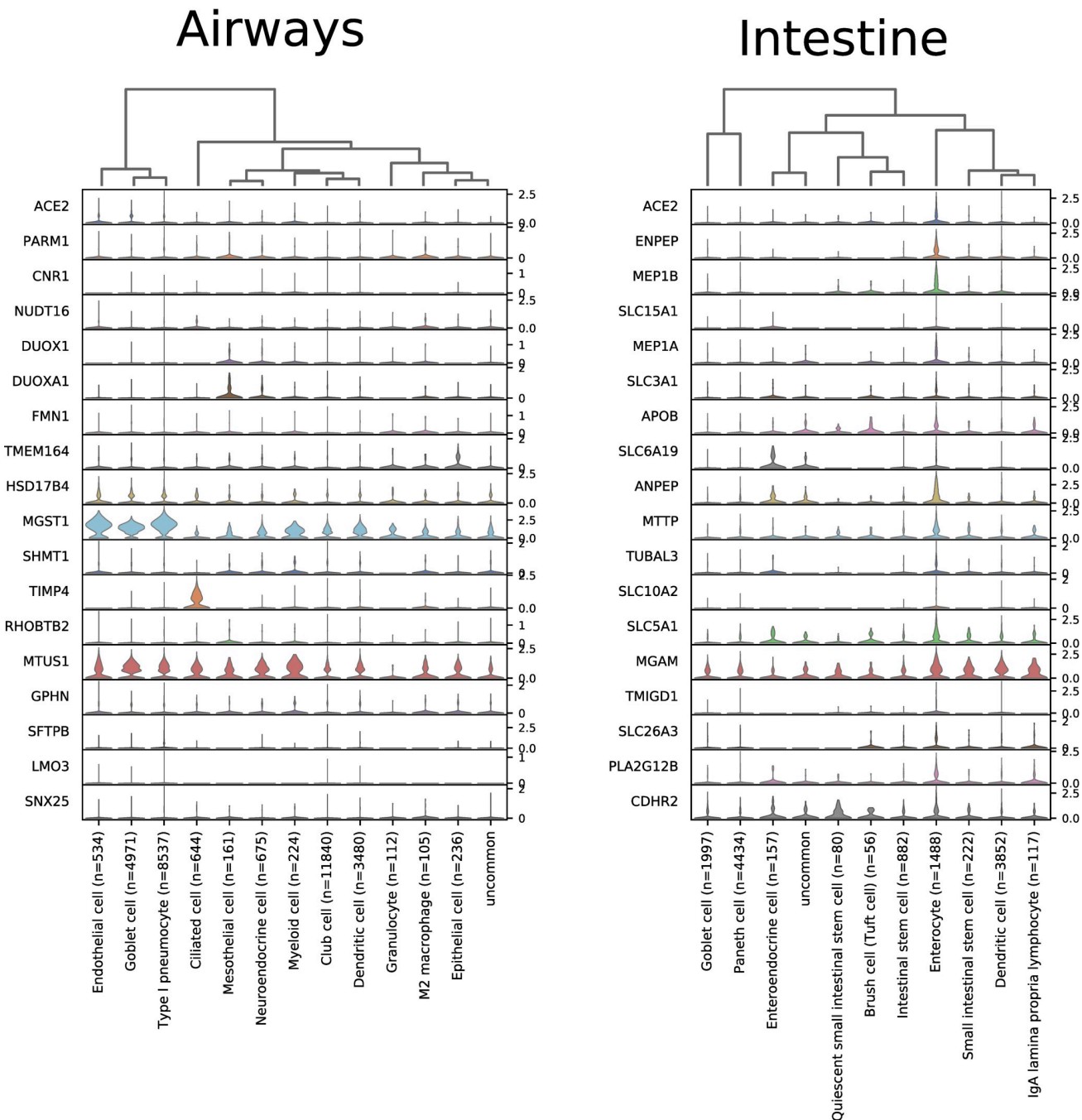

**Fig 10. Expression of genes most highly correlated with *ACE2* in single cell datasets of trachea and intestinal epithelia.** Trachea expression data is taken from GSE103354 [51] and intestinal epithelia data is derived from GSE92332 [52]. Visualized in ScanPy [54].

of the respiratory tract. Our analysis of *ACE2* expression in the human lung show low levels of expression in all cell types, with arterial vascular endothelial cells achieving the highest overall ratio of just ~2.5%. The present results based on mouse tracheal dataset suggested that *ACE2* mRNA is predominantly expressed in the club cells, goblet cells, and ciliated epithelial cells, and at significantly higher frequency than found in the lung. The mouse dataset used in our study contained no secretory3 cells, which Lukassen and colleagues recently reported to

**Table 2. Gene ontology annotation results for the processes associated with genes strongly coexpressed (≥0.5) with *ACE2* across all tissues in GTEx dataset.** Calculated with GProfiler Python library [41].

| Source | Native | Process | p-value |
|---|---|---|---|
| GO:BP | GO:0001525 | angiogenesis | 1.797E-08 |
| GO:BP | GO:0048514 | blood vessel morphogenesis | 4.611E-08 |
| GO:BP | GO:0001944 | vasculature development | 1.893E-07 |
| GO:BP | GO:0072358 | cardiovascular system development | 2.365E-07 |
| GO:BP | GO:0001568 | blood vessel development | 4.746E-07 |
| GO:BP | GO:0035239 | tube morphogenesis | 5.907E-07 |
| GO:BP | GO:0072359 | circulatory system development | 6.641E-07 |
| GO:BP | GO:0048646 | anatomical structure formation involved in morphogenesis | 1.885E-05 |
| GO:BP | GO:0035295 | tube development | 2.900E-05 |
| GO:BP | GO:1901342 | regulation of vasculature development | 0.025 |
| GO:BP | GO:0008217 | regulation of blood pressure | 0.035 |
| GO:BP | GO:0045765 | regulation of angiogenesis | 0.009 |
| GO:CC | GO:0071944 | cell periphery | 0.001 |
| GO:CC | GO:0009986 | cell surface | 0.002 |
| GO:CC | GO:0005886 | plasma membrane | 0.013 |
| GO:CC | GO:0046930 | pore complex | 0.003 |
| HP | HP:0005381 | recurrent meningococcal disease | 0.004 |
| HP | HP:0005430 | recurrent Neisserial infections | 0.007 |
| HP | HP:0100601 | eclampsia | 0.021 |
| KEGG | KEGG:04610 | complement and coagulation cascades | 0.002 |
| KEGG | KEGG:04923 | regulation of lipolysis in adipocytes | 0.037 |
| WP | WP:WP558 | complement and coagulation cascades | 0.011 |

express the highest levels of *ACE2* mRNA along the human respiratory tract [65]. Another study reported positive expression in the type AT2 pneumocytes [66], which is in line with the results of Lukassen et al. [65], but only a few cells appeared positive. A third study based on single cell expression data demonstrated the strongest positive signal in the lung AT2 cells, while other cells including AT1 cells, club cells, ciliated cells, and macrophages showed weaker expression [67]. A fourth single cell expression analysis using Gene Expression Omnibus (GEO) database recently demonstrated *ACE2*-positive signal in 1% of AT2 cells and in 2% of respiratory tract epithelial cells [68]. This correlates with our own findings for lung. For comparison about 30% of ileal epithelial cells were *ACE2*-positive, and 44% of enterocytes in small intestine of mouse. Immunohistochemical analysis of mouse tissues has shown positive signal in the club cells, AT2 cells, endothelial cells, and smooth muscle cells [69]. In spite of the obvious discrepancies between different datasets, that highlights the need for large numbers of thoroughly characterized cells for single cell RNA-Seq analyses, we can now make some conclusions of the expression of *ACE2* mRNA in the respiratory tract. First, *ACE2* is positively though weakly expressed in the AT2 cells of the lung and less so in AT1 cells. Second, *ACE2* also shows a weak positive signal, but at significantly higher proportions of cells, in several other cell types of the trachea, including goblet cells, club cells, and ciliated cells. Third, based on the findings of Lukassen et al. [65] secretory3 cells, a transient cell type of the bronchial tree, may express the highest levels of *ACE2*. These ACE2-positive cell types may represent the main host cells for SARS-CoV-2 along the whole respiratory tract. However, the median percentage of ACE-expressing secretory3 cells in the study was less than 6%, significantly less than that observed in club (16.62%), goblet (13.84%), and ciliated (6.63%) cells of trachea we have identified in the GSE103354 dataset.

Goblet cells, ciliated epithelial cells, and club cells are considered important cell types for the protection of airway mucosa. Lukassen and coworkers [65] described secretory3 cells as intermediate cells between goblet, ciliated, and club cells. If SARS-coronaviruses predominantly attack these cells, locating along the airway segments including the trachea, bronchi, and bronchioles until the last segment that is the respiratory bronchioles, it would be obvious that physiological protective mechanisms are severely affected. Defective mucosal protection and inefficient removal of pathogens due to viral infection may contribute to onset of severe bilateral pneumonia that is common for SARS-diseases [70]. This pathogenic mechanism is supported by previous findings, showing that early disease is manifested as a bronchiolar disease with respiratory epithelial cell necrosis, loss of cilia, squamous cell metaplasia, and intrabronchiolar fibrin deposits [32]. In fact, it has been suggested that early diffuse damage as a result of SARS-CoV-1 infection may actually initiate at the level of the respiratory bronchioles [71, 72].

Our findings confirm that the respiratory tract tissues have quite limited expression levels of *ACE2* compared to several other tissues that show much more prominent signal. Because ACE2 is highly expressed in the intestine [29], as also confirmed by our bioinformatics study, it would be obvious to predict that both SARS-CoV-1 and -2 infections cause significant gastrointestinal pathology and symptoms including diarrhea. Interestingly, the patients with COVID-19 have reported less gastrointestinal symptoms than the SARS-CoV-1-infected patients [22, 30]. The pathophysiological basis for this phenomenon is not understood at this point, and thus further investigations on this topic are warranted.

When we initiated the present study, we hypothesized that understanding better the transcriptional regulation of the *ACE2* gene might help to explain the peculiar distribution pattern of ACE2 in tissues. Since upregulation of ACE2 would reflect an increased number of SARS-coronavirus receptors on cell surfaces, it could possibly help us to understand the mechanisms why certain patients (males more than females, old more than young, smokers more than non-smokers) are more susceptible to the most detrimental effects of the COVID-19 infection. In our study, the signals for *ACE2* mRNA in the lung specimens did not vary much in different age groups nor did they show significant differences between males and females, which is in line with previous findings [65]. Therefore, different expression levels of lung ACE2 may not explain the variable outcome of the disease concerning age groups and genders. Importantly, our studies on this aspect were performed using whole tissue RNA-Seq values, and at least one other analysis using single-cell RNA-Seq data has identified changes in *ACE2* expression associated age, sex, and smoking status for various cell types [73]. Specifically, they have found *ACE2* expression to increase with age in basal and multiciliated cells, and higher expression for males in airway secretory cells and alveolar AT2 cells. Additionally, a study of *ACE2* expression in nasal epithelium (not included in GTEx dataset) showed lowest levels in young children (<10) with increasing values in later age groups [74]. It has been recently discussed that different innate and adaptive immune responses related to both age and gender may contribute to variable outcome of severe viral diseases [35]. It is clearly one major research area to be followed regarding COVID-19 infection.

To investigate the transcriptional regulation of the *ACE2* gene we made predictions for the binding sites of transcription factors within the proximal promoter region of the intestine-specific and lung-specific human *ACE2* transcript promoters. Our findings introduced several putative binding sites in the *ACE2* promoter for known transcription factors, which showed high levels of coexpression with ACE2 in several tissues including the ileum, colon, and kidney. The identified transcription factors could represent potential candidate target molecules which regulate *ACE2* expression. Two of our predictions, for HNF1A and HNF1B, have been previously identified experimentally to drive ACE2 expression in pancreatic islet cells and

insulinoma cells, respectively [49]. Later work by the same group has shown that our prediction of FOXA binding sites in the *ACE2* promoter are also likely correct [75]. It is of interest that *ACE2* might be regulated by oxygen status. Zhang and coworkers previously demonstrated that *ACE2* mRNA and protein levels increased during the early stages of hypoxia and decreased to near-baseline levels at later stages after hypoxia inducible factor (HIF)-1α accumulation [76]. Based on these findings *ACE2* has been listed as a HIF1α-target gene [77], although it does not follow the typical HIF1α regulated expression pattern, nor is there any predicted HIF1α binding site in our analyses. However, HNF1B has been identified as upregulated in hypoxia in kidney, independent of HIF1α [78], and in hypoxic embryonic stem cells HIF1α has been shown to increase expression of transcription factors TCF7/LEF1 (predicted to bind the promoter of the lung-specific ACE2 transcript) through Wnt/β-catenin signaling [79].

Recent work has shown that ACE2 expression is stimulated by interferon alpha (IFN-α) *in vitro* and computationally identified evidence for STAT1, STAT3, IRF8, and IRF1 TFBSs in the *ACE2* promoter [80]. Another recent study has identified correlations of expression between *ACE2* and other interferon stimulated genes [81]. Our analysis produced several putative binding sites for interferon-stimulation mediating TF genes, proximal to the TSSs of the intestine-specific (STAT1, IRF8, and IRF9) and lung-specific transcript (IRF8). The findings of these studies, and our own, potentially reveal a scenario where SARS-CoV-2 infection itself may induce expression of ACE2 and thus provide a self-perpetuating route of increased cellular infection. This could explain how such low overall ACE2 expression in normal lungs translates into a fatal disease state. However, a series of recent pre-print articles have identified a previously unreported novel *ACE2* transcript with a transcription start site occurring between exons 9 and 10 of the canonical *ACE2* gene. Alternatively, this truncated transcript has been named *LTR16A1-ACE2* [56], *delta-ACE2* [57], or simply 'short *ACE2*' [58]. In all cases the authors have concluded that this new transcript is more strongly stimulated by interferon, though their results differ on whether transcription of the long form is [58] or is not [56, 57] interferon-related. Importantly, a protein resulting from this transcript would lack the 356 amino acids of the first 9 exons, and thus not contain the known SARS-CoV-2 binding domains, and all three studies have hypothesized that the short form ACE2 would not likely be a point of viral entry. One study was unsuccessful in attempting to produce a viable protein from the short transcript [56], while others [57, 58] identified peptides indicative of the short ACE2 protein in data from mass-spec analysis of cancer samples (ovary, colon, breast) from The Cancer Genome Atlas (TCGA). In addition, Blume and coworkers were able to use Western blotting of an ACE2 antibody targeting the C-terminal domain to identify appropriately sized bands (~50kDa), expected of the short form, in nasal epithelial cells and bronchial epithelial cells [58]. All three studies found that the ratio of expression of the short form to long form is highest in nose and mouth and reduces as you progress down the airways and digestive tracts.

Ng et al. in [56] have identified a long terminal repeat (LTR) (LTR16A1) occurring in intron 9 of *ACE2*, comprising most of the first exon of short-form *ACE2*, which they posit may be related to the observed interferon reactivity. Our TFBS analysis of LTR16A1 and the promoter of the short *ACE2* transcript revealed putative binding sites for multiple IRF proteins and a STAT1:STAT2 dimer immediately upstream (<50 bp) of the newfound TSS and LTR16A1 (Fig 9), and STAT protein binding sites further upstream. While all three studies promote this new short form of *ACE2* as more strongly upregulated by interferon, two [57, 58] also contain results that show that long form *ACE2* is itself also upregulated, to a lesser extent, in presence of interferon or infection.

These three manuscripts represent the leading edge of, and first forays into, our understanding of *ACE2* expression in response to interferon. Accordingly, while there are broad strokes of agreement, there is not consensus on all points. Peer-review of the results needs to be first performed, and subsequent study is needed to validate and expand these new findings. In summary, at present, there is evidence that both short and long forms of *ACE2* are upregulated by interferon, however the short form appears to be more strongly upregulated, appears to be expressed at higher levels in nose and mouth, and lacks the domains currently understood to bind the SARS-CoV-2 spike protein. Our results show significant predictions for interferon response elements in the proximal promoters of all three *ACE2* transcripts. Further deepening the importance of understanding the role of interferon in COVID-19 is a new study showing that the presence of auto-antibodies against either or both of interferon alpha (IFNα) and interferon omega (IFNω) were rare in healthy, asymptomatic, or mild SARS-CoV-2 infection, but over-represented among patients with life-threatening COVID-19 pneumonia [82]. The full story of the regulation of ACE2 expression remains an enigma and there appear to be many factors involved. One limitation of our study is that it is focused on mRNA expression and transcriptional regulation only. There may exist factors which function at posttranscriptional level. Indeed, Srivastava and colleagues recently demonstrated that SARS-CoV-2 infection induces alterations in the post-transcriptional regulatory networks in human tissues through the function of RNA binding proteins and micro-RNAs [83]. There has been clinical concern that the use of ACE inhibitors and angiotensin receptor blockers could increase the expression of ACE2 and increase patient susceptibility to viral host cell entry [84, 85]. Previous studies have suggested that both ACE inhibitor and angiotensin II receptor type I antagonist therapies increase *ACE2* mRNA expression in rat heart [86]. There has also been some evidence in humans showing increased expression of ACE2 in the heart, brain, and even in urine after treatment with angiotensin receptor blockers [84]. Since these drugs are widely used for treatment of hypertension and heart failure, it would be important to determine in COVID-19 patients whether these medications have any significant effects on symptoms or outcome of the disease.

Gene ontology investigations revealed interesting novel data on potential physiological roles of ACE2. The five most significant gene ontology terms included angiogenesis, blood vessel morphogenesis, vasculature development, cardiovascular system development, and blood vessel development. Angiotensin-(1–7) is a direct product of ACE2, and through binding with the Mas receptor has been shown to advance angiogenesis in injured cardiac tissue (myocardial infarction), by increasing expression of VEGF-D and MMP-9 [87], and in stroke [88]. Other studies have suggested that ACE2, either by reducing angiotensin II or through activities of the ACE2/angiotensin-(1–7)/MasR axis, may be negatively associated with angiogenesis in various cancers [81, 89–91]. It also appears to play a role in angiogenesis in uterus during pregnancy [92]. Our study of scRNA-Seq data from human lung showed that the arterial and venous vascular endothelial cell types had the highest ratios of ACE2-expressing cells, first and third highest, respectively. In another study, ACE2 expression was detected in blood vessels [27], while a recent study showed that SARS-CoV-2 is capable of directly infecting blood vessel cells [12]. Endothelial ACE2 expression may be linked to clotting and multi-organ dysfunction reported in many patients with COVID-19 [93]. Our GO analysis provided evidence that ACE2 is involved in the KEGG pathway 'complement and coagulation cascades'. Indeed, patients with severe COVID-19 often present with coagulation abnormalities that mimic other known systemic coagulopathies, such as disseminated intravascular coagulation (DIC) or thrombotic microangiopathy, but COVID-19 has its own distinct features [94]. Based on the present finding, angiogenesis/blood vessel morphogenesis may be considered another putative function for ACE2 in addition to its classical role as the key angiotensin-(1–7) forming enzyme [95].

## Conclusions

Our bioinformatics study confirmed the low expression of *ACE2* in the respiratory tract. In lung it was lowest of all, while significantly higher in the trachea. Bulk RNA-Seq analyses indicated the highest expression levels in the small intestine, colon, testis, and kidney. In the human lung scRNA-Seq dataset, the strongest positive signals for *ACE2* mRNA were observed in vascular endothelial cells, goblet cells, ciliated cells, and AT2 and AT1 pneumocytes. In the mouse trachea dataset, positive signals were most common in club cells, goblet cells and ciliated epithelial cells. The results suggest that SARS-CoV infection may target the cell types that are important for the protection of airway mucosa and their damage may lead to deterioration of epithelial cell function, finally leading to a more severe lung disease with accumulation of alveolar exudate and inflammatory cells and lung edema, the signs of pneumonia recently described in the lung specimens of two patients with COVID-19 infection [96]. Gene ontology analysis based on expression in all tissues suggested that ACE2 is involved in angiogenesis/blood vessel morphogenesis processes in addition to its classical function in renin-angiotensin system. Many findings reported here have not yet been verified *in vivo* or *in vitro*. Therefore, the validity of the bioinformatics results needs to be verified by future experimental research.

## Supporting information

**S1 Fig.**
(PDF)

**S1 Table.**
(XLSX)

**S2 Table.**
(XLSX)

**S3 Table.**
(XLSX)

**S4 Table.**
(XLSX)

**S5 Table.**
(XLSX)

**S6 Table.**
(XLSX)

**S7 Table.**
(XLSX)

**S8 Table.**
(XLSX)

## Author Contributions

**Conceptualization:** Harlan Barker, Seppo Parkkila.

**Data curation:** Harlan Barker.

**Formal analysis:** Harlan Barker, Seppo Parkkila.

**Funding acquisition:** Seppo Parkkila.

**Investigation:** Harlan Barker, Seppo Parkkila.

**Methodology:** Harlan Barker.

**Project administration:** Seppo Parkkila.

**Resources:** Harlan Barker, Seppo Parkkila.

**Software:** Harlan Barker.

**Supervision:** Seppo Parkkila.

**Validation:** Harlan Barker.

**Visualization:** Harlan Barker, Seppo Parkkila.

**Writing – original draft:** Harlan Barker, Seppo Parkkila.

**Writing – review & editing:** Harlan Barker, Seppo Parkkila.

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
