## [Decision Letter · Decision Letter 0]

21 Jul 2020

PONE-D-20-16903

Bioinformatic characterization of angiotensin-converting enzyme 2, the entry receptor for SARS-CoV-2

PLOS ONE

Dear Dr. Parkkila,

Thank you for submitting your manuscript to PLOS ONE. After careful consideration, we feel that it has merit but does not fully meet PLOS ONE’s publication criteria as it currently stands. Therefore, we invite you to submit a revised version of the manuscript that addresses the points raised during the review process.

We look forward to receiving your revised manuscript.

Kind regards,

Michael Bader

Academic Editor

PLOS ONE

Journal Requirements:

Reviewers' comments:

Reviewer's Responses to Questions

**Comments to the Author**

1. Is the manuscript technically sound, and do the data support the conclusions?

Reviewer #1: Yes

Reviewer #2: Partly

Reviewer #3: Partly

2. Has the statistical analysis been performed appropriately and rigorously? 

Reviewer #1: Yes

Reviewer #2: Yes

Reviewer #3: Yes

3. Have the authors made all data underlying the findings in their manuscript fully available?

Reviewer #1: Yes

Reviewer #2: Yes

Reviewer #3: Yes

4. Is the manuscript presented in an intelligible fashion and written in standard English?

Reviewer #1: Yes

Reviewer #2: Yes

Reviewer #3: Yes

5. Review Comments to the Author

Reviewer #1: 1. Figure 3 & 4: is not clear, the image quality is not good and different cell types is not readable.

2. What is the reason behind using both human and mouse tissues for the study? was there any difference between the ACE2 expression in mouse and human in different organs ? Please comment.

3. The results in the current paper with respect to ACE2 mRNA expression was contrasting with Lukassen and colleague, as they claimed high expression of ACE2 in human respiratory tract, which is the reason behind COVID19 patients experiencing pneumonia. Please explain.

4. Page 21, line 56-71, the authors conclude the high expression of ACE2 in Club, goblet and ciliated cells of trachea, which can cause respiratory distress, but the overall expression of ACE2 in figure 1 is highest in small intestine. What could be the reason behind the COVID19 patients not experiencing the gastrointestinal symptom as seen in SARs-CoV-1. The Club, goblet and ciliated cells are also expressed in the intestine. What could be the reason behind COVID19 infection severely affecting lungs?

5. The authors concluded, no difference in ACE2 expression was observed between the age groups and genders. Please reconcile with COVID19 reports adversely infecting older population and men.

6. The authors identified the possible transcription factor binding site, which drive ACE2 expression in lung and intestine. The figure 9 showed MGST1, MTUS1, and MGAM could be potential site, which needs to be discussed further. As I see the discussion part is highlighting the findings from other papers.

7. Expression of ACE2 in the current study does not match original reports of ACE2 expression (Harmer et al. 2002). Please explain.

Reviewer #2: In this paper, a secondary data analysis on human and mouse data sets was performed in regard to ACE2 mRNA expression using bioinformatics tools. The authors attempted to determine cell-specific expression of ACE2 mRNA in a number of organs that could likely be viewed as for SARS-Cov-2 virus entry ports. Prediction of transcription factor binding sites, gene ontology and sex-comparative analysis were also performed. The authors hypothesized that using these bioinformatic tools one could predict functions and transcriptional regulation of ACE2 yet not all approaches had been shown to be validated yet (see below).

The introduction clearly delineates the rationale for the studies performed. The Discussion is well structured and gives a logical summary of the presented findings. Less organized is the Results section. The data presented here also reveals some rigor deficiencies which makes the paper less convincing. Also, the approach has to be critically viewed by cautioning that other levels than transcriptional could be operative in regulating expression of ACE2.

In summary, the paper provides some interesting data and conclusions. However, there are deficiencies that need to be addressed.

Specific comments:

• Many studies, and also the Protein Atlas website, a tool the authors used themselves, have reported high ACE2 protein expression in the kidney yet the ACE2 mRNA expression in the kidney was reported here as relatively low. By unsophisticated comparison, there is a discrepancy that might suggest a level of regulation beyond the transcription. Moreover, there have been reports on post-transcriptional regulation of ACE2, e.g. in diabetes. It should be stated that one of the limitations of the approach used is that it is based on mRNA expression and transcriptional regulation only and that other levels of ACE2 regulation could be effective as well.

• The novel computational tool that has been recently developed by the first author of this article has not been published. Yet the authors use it to predict transcriptional regulation. Has this tool been validated in any way? How this tool works and how was it verified that the tool really works. If the tool has not been published and no data on its validation has been provided, should the reader just take it for granted? Some information has to be given on this or, if already available, the appropriate reference cited.

• Given that many of the findings have not yet been verified in vivo or in vitro, it would be appropriate to mention that the validity for the majority of the findings presented here would need to be verified by future research.

• In this paper, a secondary data analysis on human and mouse data set studies was performed. It takes a bit effort for the reader to find out what was studied. Given that various sources that were used for this, it would be useful to give some overview in form of a table or so: the source of data set, species, what was studied.

• There is insufficient information how the authors performed evaluation of the images that can be accessed from the Protein Atlas. How many images assessed, what magnification etc.

• Lanes 119 and 125 - Information is missing on what is GTEx database and where can be found.

• Is the expression of ACE2 in the kidney moderate (Abstract, Figure 2) of one of the highest (Discussion, Conclusions)?

Reviewer #3: In their manuscript, Barker and Parkkila investigate the expression and transcriptional regulation of ACE2, the receptor of SARS-CoV-2, through the analysis of existing datasets. Specifically, they determine the tissue and cell type specificity, investigate the covariates age and gender, and identify putative transcriptional regulators of ACE2 as well as genes with highly correlated expression patterns.

The manuscript is generally well written and nicely merges different layers of information on the expression and potential role of ACE2. The description of statistical tests is thorough, and appropriate tests are used for the gender and correlation analyses. For the analysis of age dependency, the one-way ANOVA implemented in scipy.stats is not an optimal choice as the ranking information of the age bins is lost. Ideally, the authors should try to work with the raw values if available and perform a regression fit here. Looking at the box plots, this should work reasonably well even with simple linear regression for most tissues, with the exception of kidney. Parts of the manuscript would benefit from a more detailed description of the approach taken, and some results contradict prior literature which would need to be addressed. Furthermore, as ACE2 is discussed specifically in its function as SARS-CoV-2 receptor, the manuscript could benefit from further analyses geared more towards ACE2 regulation in the context of an infection. I would not consider the latter point to be strictly necessary, provided the authors clearly state that their analysis is a representation of the physiological expression pattern and may not fully hold during a SARS-CoV-2 infection, but if it can be done quickly it would be worth considering.

Specific comments:

• It is sometimes unclear which dataset an analysis was performed on. In general, a structured list of datasets including links to the resources in the Methods section or as a supplementary table would be a great help.

• For the TFBS analysis, I’m not sure what the tissues in parentheses (e.g. lines 325/326) indicate. Was a correlation observed in these tissues (if yes, was the dataset bulk or single-cell)?

• It remains somewhat unclear how TFBS footprinting works. Based on the GitHub repository it mostly seems to aggregate PWM scores with experimental measures for TF occupancy. Given that there is no paper or preprint yet, the results presented in this manuscript should be explained in more detail (e.g., what data is the ATAC-Seq track based on? Is it lung or some other organ/aggregate database?). In addition, some corroborating evidence, or a rationale for why this is not needed, i.e. the data shown can directly be obtained from a database (which?), should be provided.

• ACE2 has been described as an interferon responsive gene. This could have implications when transferring the results presented here to the situation in COVID-19, as no major interferon signaling would be expected in healthy tissues. This could also help explain the disparity between the severity of COVID-19 and the lack of ACE2 expression in the lung beyond the initial infection. Several STAT/IRF TFs appear to have a significant hit according to the PWM scoring (and in the case of STAT1 for combined affinity). Since this interferon-responsiveness has now been found by several studies, it should be discussed, or directly addressed through inclusion of an infected dataset. Ideally that would be one of the COVID-19 datasets slowly becoming available, but if the question is just about interferon response, other infections such as influenza should work as well.

• The findings on the influence of age and gender contradict some existing literature on the subject (Muus et al., bioRxiv 2020; Bunjavanich et al., JAMA 2020; indirectly as they measured activity rather than expression: Soro-Paavonen et al., Journal of Hypertension 2012; Fernández-Atucha et al., Biology of Sex Differences 2017). While I wouldn’t necessarily consider the issue settled yet given the low expression of ACE2 and associated high levels of noise, these disparities should definitely be addressed.

• As ACE2 is rather strongly expressed in the vasculature, I would have been surprised to not see angiogenesis pop up as enriched pathway. While I wouldn’t discount the possibility that ACE2 does indeed play a role in this process, it will be hard to reliably make this point based on the datasets included in this manuscript. While the authors could flesh out their analysis with angiogenesis datasets, ideally showing a differential and stage-specific expression of ACE2 in the process on the RNA and/or protein level. There is a paper describing an ACE2 knockout mouse (Crackower et al., Nature 2002), but the phenotype they describe seems to hint more towards the “canonical” functions of ACE2.

• Fernández-Atucha et al. describe an age-dependent effect on ACE2 activity in females only (by linear regression). It might be worth looking at the interaction term rather than performing a one-way ANOVA here.

Minor comments:

The citation of the primary publication of the human protein atlas appears to be missing (see https://www.proteinatlas.org/about/licence).

6. PLOS authors have the option to publish the peer review history of their article (what does this mean?). If published, this will include your full peer review and any attached files.

Reviewer #1: No

Reviewer #2: No

Reviewer #3: No

---

## [Author Response · Author response to Decision Letter 0]

4 Sep 2020

We thank the reviewers for their valuable comments. Our responses are in the attachment.

---

## [Decision Letter · Decision Letter 1]

18 Sep 2020

PONE-D-20-16903R1

Bioinformatic characterization of angiotensin-converting enzyme 2, the entry receptor for SARS-CoV-2

PLOS ONE

Dear Dr. Parkkila,

Thank you for submitting your manuscript to PLOS ONE. After careful consideration, we feel that it has merit but does not fully meet PLOS ONE’s publication criteria as it currently stands. Therefore, we invite you to submit a revised version of the manuscript that addresses the point raised by reviewer 3 refering to the newly discovered isoform of human ACE2.

We look forward to receiving your revised manuscript.

Kind regards,

Michael Bader

Academic Editor

PLOS ONE

Reviewers' comments:

Reviewer's Responses to Questions

**Comments to the Author**

1. If the authors have adequately addressed your comments raised in a previous round of review and you feel that this manuscript is now acceptable for publication, you may indicate that here to bypass the “Comments to the Author” section, enter your conflict of interest statement in the “Confidential to Editor” section, and submit your "Accept" recommendation.

Reviewer #2: All comments have been addressed

Reviewer #3: All comments have been addressed

2. Is the manuscript technically sound, and do the data support the conclusions?

Reviewer #2: Yes

Reviewer #3: Yes

3. Has the statistical analysis been performed appropriately and rigorously? 

Reviewer #2: Yes

Reviewer #3: Yes

4. Have the authors made all data underlying the findings in their manuscript fully available?

Reviewer #2: Yes

Reviewer #3: Yes

5. Is the manuscript presented in an intelligible fashion and written in standard English?

Reviewer #2: Yes

Reviewer #3: Yes

6. Review Comments to the Author

Reviewer #2: (No Response)

Reviewer #3: The authors have completely and thoroughly addressed my concerns, however, I cannot recommend acceptance of the article yet.

I am deeply sorry for raising another issue that wasn't included in the first round of revisions, but days after the decision on the previous version of the manuscript, three preprints (https://www.biorxiv.org/content/10.1101/2020.07.24.219139v1, https://www.biorxiv.org/content/10.1101/2020.07.19.210955v1, https://www.biorxiv.org/content/10.1101/2020.07.31.230870v1) were uploaded to bioRxiv that shed more light on the interferon-dependent regulation of ACE2, independently identifying a previously unknown, likely non-functional isoform of ACE2 that is strongly interferon-responsive. This isoform stems from an alternative promoter, leading to a 5' truncation that is not picked up by 3' (single-cell) RNA-Seq. As this is highly relevant to the question of regulation of ACE2 during COVID-19, I would strongly urge the authors to add an analysis of this alternative promoter. Furthermore, it might be worth looking at GTEx exon-level data at least for tissues where some sort of differential expression is observed, or restrict the comparison to the 5' exons to capture just the full-length isoform. If no major differences are observed, this information could be moved to a supplementary figure.

For this reason, I recommend a major revision. Please note that this is not reflective of the general quality of the work: Had this additional information not been made available in the meantime, I would have happily recommended acceptance of the article as-is.

Minor comment:

The quality of some figures is still rather bad. I think I had a similar issue with Editorial Manager in the past, where the uploaded high quality images were compressed too strongly. If that is the case, everything is fine. If the images in the reviewer file are reflective of the actual quality though, I would urge a shift to vector graphics or at least higher resolution images.

7. PLOS authors have the option to publish the peer review history of their article (what does this mean?). If published, this will include your full peer review and any attached files.

Reviewer #2: No

Reviewer #3: No

---

## [Author Response · Author response to Decision Letter 1]

28 Sep 2020

Reviewer #3:

Reviewer #3: The authors have completely and thoroughly addressed my concerns, however, I cannot recommend acceptance of the article yet. I am deeply sorry for raising another issue that wasn't included in the first round of revisions, but days after the decision on the previous version of the manuscript, three preprints (https://www.biorxiv.org/content/10.1101/2020.07.24.219139v1, https://www.biorxiv.org/content/10.1101/2020.07.19.210955v1, https://www.biorxiv.org/content/10.1101/2020.07.31.230870v1) were uploaded to bioRxiv that shed more light on the interferon-dependent regulation of ACE2, independently identifying a previously unknown, likely non-functional isoform of ACE2 that is strongly interferon-responsive. This isoform stems from an alternative promoter, leading to a 5' truncation that is not picked up by 3' (single-cell) RNA-Seq. As this is highly relevant to the question of regulation of ACE2 during COVID-19, I would strongly urge the authors to add an analysis of this alternative promoter. Furthermore, it might be worth looking at GTEx exon-level data at least for tissues where some sort of differential expression is observed, or restrict the comparison to the 5' exons to capture just the full-length isoform. If no major differences are observed, this information could be moved to a supplementary figure.

For this reason, I recommend a major revision. Please note that this is not reflective of the general quality of the work: Had this additional information not been made available in the meantime, I would have happily recommended acceptance of the article as-is.

OUR RESPONSE: We thank the reviewer for this valuable comment. We have revised the manuscript according to the new information. We have added the following texts and other materials (lines correspond to the version with ”track changes”:

Abstract, lines 37-38

Methods, lines 165-166

Results, lines 360-375

Discussion, lines 125-162

New Figure 9 has been added

References 56, 57, 58, and 82 have been added

Supplementary Table 5 has been revised

MINOR COMMENT: The quality of some figures is still rather bad. I think I had a similar issue with Editorial Manager in the past, where the uploaded high quality images were compressed too strongly. If that is the case, everything is fine. If the images in the reviewer file are reflective of the actual quality though, I would urge a shift to vector graphics or at least higher resolution images.

OUR RESPONSE: The figure quality has been poor because of the automated compression for the reviewer version of the manuscript. It shouldn´t be an issue for the final version of the paper. 

OTHER MINOR CHANGES:

Gene abbreviations are written in Italic.

Supplementary Figure 1 has been updated to include STAT dimers which had been accidentally left out in the first analysis.

The number of COVID cases has been updated (lines 56-57).

---

## [Editor Report · Decision Letter 2]

1 Oct 2020

Bioinformatic characterization of angiotensin-converting enzyme 2, the entry receptor for SARS-CoV-2

PONE-D-20-16903R2

Dear Dr. Parkkila,

We’re pleased to inform you that your manuscript has been judged scientifically suitable for publication and will be formally accepted for publication once it meets all outstanding technical requirements.

Kind regards,

Michael Bader

Academic Editor

PLOS ONE
---

## [Editor Report · Acceptance letter]

5 Oct 2020

PONE-D-20-16903R2 

Bioinformatic characterization of angiotensin-converting enzyme 2, the entry receptor for SARS-CoV-2 

Dear Dr. Parkkila:

I'm pleased to inform you that your manuscript has been deemed suitable for publication in PLOS ONE. Congratulations! Your manuscript is now with our production department. 

Kind regards, 

on behalf of

Prof. Michael Bader 

Academic Editor

PLOS ONE